# PaCA: Partial Connection Adaptation for Efficient Fine-Tuning

**Sunghyeon Woo, Sol Namkung, Sunwoo Lee, Inho Jeong, Beomseok Kim, Dongsuk Jeon**
Seoul National University
{wsh0917, djeon1}@snu.ac.kr

## Abstract

Prior parameter-efficient fine-tuning (PEFT) algorithms reduce memory usage and computational costs of fine-tuning large neural network models by training only a few additional adapter parameters, rather than the entire model. However, the reduction in computational costs due to PEFT does not necessarily translate to a reduction in training time; although the computational costs of the adapter layers are much smaller than the pretrained layers, it is well known that those two types of layers are processed sequentially on GPUs, resulting in significant latency overhead. LoRA and its variants avoid this latency overhead by merging the low-rank adapter matrices with the pretrained weights during inference. However, those layers cannot be merged during training since the pretrained weights must remain frozen while the low-rank adapter matrices are updated continuously over the course of training. Furthermore, LoRA and its variants do not reduce activation memory, as the first low-rank adapter matrix still requires the input activations to the pretrained weights to compute weight gradients. To mitigate this issue, we propose **Pa**rtial Connection **A**daptation (**PaCA**), which fine-tunes randomly selected partial connections within the pretrained weights instead of introducing adapter layers in the model. PaCA not only enhances training speed by eliminating the time overhead due to the sequential processing of the adapter and pretrained layers but also reduces activation memory since only partial activations, rather than full activations, need to be stored for gradient computation. Compared to LoRA, PaCA reduces training time by 22% and total memory usage by 16%, while maintaining comparable accuracy across various fine-tuning scenarios, such as fine-tuning on the MMLU dataset and instruction tuning on the Oasst1 dataset. PaCA can also be combined with quantization, enabling the fine-tuning of large models such as LLaMA3.1-70B. In addition, PaCA enables training with 23% longer sequence and improves throughput by 16% on both NVIDIA A100 GPU and INTEL Gaudi2 HPU compared to LoRA. The code is available at https://github.com/WooSunghyeon/paca.

## 1 Introduction

Following the scaling laws (Kaplan et al., 2020; Hoffmann et al., 2022), the size of language models based on the transformer architecture (Vaswani et al., 2017) has grown significantly in recent years. Large Language Models (LLMs) such as GPT4 (OpenAI, 2023) and LLaMA 3 (Dubey et al., 2024) have achieved remarkable abilities across a wide range of general tasks. Furthermore, the capabilities of LLMs can be refined for specific purposes, either by creating models specialized for specific tasks through fine-tuning (Singhal et al., 2023) or by developing chatbots that better understand user queries through instruction tuning (Wei et al., 2022; Taori et al., 2023). However, fine-tuning LLMs consumes significant computational power and memory, making it impossible to perform without a large number of expensive GPUs.

Parameter-efficient fine-tuning (PEFT) (Li & Liang, 2021; Houlsby et al., 2019; He et al., 2022) is a set of methods to relieve the high costs of fine-tuning large models. Prior PEFT schemes introduce new adapter layers with significantly fewer parameters to a pretrained model and only train these newly introduced adapter layers, substantially reducing the memory needed to store gradients and optimizer states. Furthermore, PEFT can reduce the computational overhead of fine-tuning, as it

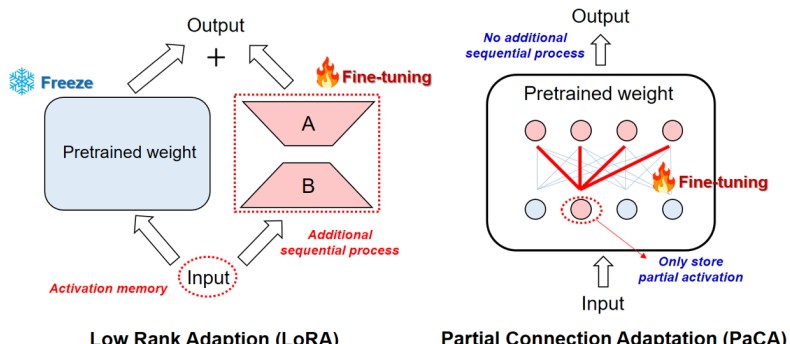

Figure 1: Overview of Partial Connections Adaptation (PaCA) algorithm.

needs to calculate the parameter gradients only for the adapter weights, rather than for all model parameters.

However, we observed that the reduction in computational cost due to PEFT does not translate into a significant decrease in actual training time. This issue arises from the fact that the adapter layers are typically processed sequentially with the pretrained layers since GPUs are generally optimized for processing one kernel at a time. This sequential processing limits the full utilization of hardware resources and incurs significant latency overhead, even though the number of FLOPs of the adapter layers is significantly smaller than that of the pretrained layers. While some software tools such as CUDA streams could be used to process the adapter layers in parallel by executing multiple kernels simultaneously, it suffers from the overhead of managing and synchronizing the streams (Wang et al., 2016; Dai et al., 2018; Han et al., 2022).

LoRA (Hu et al., 2022) and its variants (Kopiczko et al., 2024; Liu et al., 2024; Wu et al., 2024) avoid this latency overhead by merging the low-rank adapter matrices and the pretrained weights to eliminate the need for sequential processing during inference. However, this approach cannot be applied to fine-tuning since the low-rank adapter matrices need to be trained separately from the frozen pretrained weights, making the overhead from sequential processing unavoidable. Furthermore, LoRA and its variants do not reduce activation memory compared to Full-FT, since the input activations of the pretrained weights still need to be stored in memory to calculate the gradients for the first low-rank adapter matrix.

In this paper, we propose **PaCA** (**Pa**rtial **C**onnection **A**daptation), which fine-tunes randomly selected partial connections in the pretrained weights without relying on adapter layers, as depicted in Fig. 1. Unlike prior PEFT schemes, PaCA successfully reduces training time since the forward and backward operations for the pretrained weights also include those for the partial connections, eliminating the need for additional sequential processing. Furthermore, since calculating the gradients for the partial weights only requires the corresponding activations, PaCA significantly reduces activation memory usage as well. We first theoretically show that PaCA can effectively converge the loss in general neural networks. In experiments with various scenarios, PaCA demonstrates substantial reductions in both training time and memory compared to prior PEFT schemes while maintaining comparable accuracy on NVIDIA A100 GPU (Choquette et al., 2021) and Intel Gaudi2 HPU (Intel Corporation, 2023). In summary, our contributions are as follows:

- We propose PaCA, a memory-efficient PEFT algorithm that fine-tunes randomly selected partial connections within pretrianed weights without using additional adapter layers.

- We theoretically prove that PaCA can converge the loss in general neural networks.

- We experimentally show that PaCA effectively reduces memory consumption and improves training speed compared to prior PEFT algorithms across various fine-tuning scenarios on different types of GPUs.

## 2 BACKGROUND & MOTIVATION

In general, training deep neural networks involves backpropagation (Rumelhart et al., 1986), which facilitates the adaptation of the model in the direction that minimizes the loss function. The equations below show the backpropagation algorithm for a linear layer:

$$\text{Forward:} \quad \mathbf{X}_{out} = \mathbf{W}\mathbf{X}_{in} \tag{1}$$

$$\text{Backward:} \quad \nabla\mathbf{X}_{in} = \mathbf{W}^T \nabla\mathbf{X}_{out} \tag{2}$$

$$\nabla\mathbf{W} = \nabla\mathbf{X}_{out} \mathbf{X}_{in}^T \tag{3}$$

where $\mathbf{W} \in \mathbb{R}^{d_{out} \times d_{in}}$, $\mathbf{X}_{in} \in \mathbb{R}^{d_{in}}$, and $\mathbf{X}_{out} \in \mathbb{R}^{d_{out}}$ denote the weights, input activations, and output activations, respectively, with $d_{in}$ and $d_{out}$ denoting the input and output dimensions of the layer. $\nabla\mathbf{W}$ and $\nabla\mathbf{X}_{in}$ represent the weight gradients and input gradients. The forward propagation computes the output activations following Eq. 1, while the backward propagation computes the input gradients (Eq. 2) and the weight gradients (Eq. 3).

Full-FT trains all layers using backpropagation, performing the operations described in Eqs. 1-3 for each layer. Consequently, Full-FT incurs significant memory overhead due to storing the gradients and optimizer states for all parameters. To lower this overhead, various PEFT schemes have been introduced. For instance, the training scheme of LoRA (Hu et al., 2022), a representative PEFT algorithm, is represented as the equations below:

$$\text{Forward:} \quad \mathbf{X}_{out} = \mathbf{W}\mathbf{X}_{in} + \mathbf{B}(\mathbf{A}\mathbf{X}_{in}) \tag{4}$$

$$\text{Backward:} \quad \nabla\mathbf{X}_{in} = \mathbf{W}^T \nabla\mathbf{X}_{out} + \mathbf{A}^T(\mathbf{B}^T \nabla\mathbf{X}_{out}) \tag{5}$$

$$\nabla\mathbf{B} = \nabla\mathbf{X}_{out} \mathbf{X}_{mid}^T, \ \nabla\mathbf{A} = \nabla\mathbf{X}_{mid} \mathbf{X}_{in}^T \tag{6}$$

where $\mathbf{B} \in \mathbb{R}^{d_{out} \times r}$ and $\mathbf{A} \in \mathbb{R}^{r \times d_{in}}$ represent the low-rank adapter matrices in LoRA, with $r$ denoting the rank of the adapter. $\mathbf{X}_{mid} \in \mathbb{R}^r$ represents the output activations after propagating through the LoRA $\mathbf{A}$ layer (i.e., $\mathbf{X}_{mid} = \mathbf{A}\mathbf{X}_{in}$). In Eqs. 4-6, we have highlighted the computations involving adapter weights in blue. Compared to Full-FT, prior PEFT schemes introduce two key changes: 1) computations for the adapters are added in forward and backward propagations (Eqs. 4-5), and 2) only the adapters are trained, excluding the pretrained weights (Eq. 6). Since the computational cost of the adapters in PEFT is typically negligible compared to that of the pretrained layers (Li & Liang, 2021; Houlsby et al., 2019; He et al., 2022; Hu et al., 2022), PEFT can reduce the overall computational cost of training by eliminating the need to compute parameter gradients for the pretrained weights.

For more detailed analysis, we calculate FLOPs and measure training time when fine-tuning the LLaMA3-8B model using Full-FT and LoRA. Experimental results show that the operation count of LoRA is approximately 33% lower than Full-FT (Fig. 2a). However, the saving in actual training time is only 0.6%, as displayed in Fig. 2b, which is far below the expected 33% decrease. To investigate this discrepancy, we analyzed the computational cost for both forward and backward propagation, as well as the actual training time.

One interesting finding is that the time required for forward propagation in LoRA increased by 33% compared to Full-FT, despite requiring a similar number of operations, as shown in Fig. 2b. This latency overhead is due to the inefficient sequential processing of the pretrained and adapter layers, as reported by Hu et al. (2022). More specifically, the operations associated with the adapter layers are conventionally executed in a sequential manner, rather than in parallel with the pretrained layers, as GPUs are typically designed to execute a single kernel at a time. Although parallel execution of the adapter layers may be feasible using CUDA streams, which allow multiple kernels to run concurrently, these methods introduce additional overhead of resource allocation and synchronization between streams (Wang et al., 2016; Dai et al., 2018; Han et al., 2022).

This sequential processing of the adapter and pretrained layers negatively impacts hardware utilization and incurs latency overhead, despite the fact that the computational cost of the adapter layers

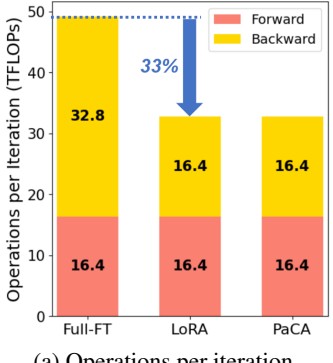
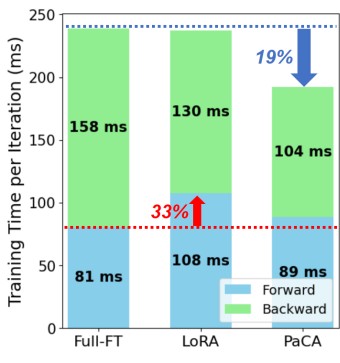

(a) Operations per iteration.           (b) Training time per iteration.

Figure 2: The number of operations (TFLOPs) and training time (ms) per iteration when training LLaMA3-8B with full-fine tuning (Full-FT) and LoRA.

accounts for only approximately 1% of that of the pretrained layers. This latency overhead could be mitigated by merging the low-rank adapter matrices into the pretrained weights during inference (Hu et al., 2022). However, during fine-tuning, where the pretrained weights must remain frozen and only the adapter weights are updated separately, such merging is not possible and the latency overhead from sequential processing remains.

Furthermore, LoRA and its variants are unable to reduce the activation memory. In Full-FT, all input activations ($\mathbf{X}_{in}$) must be stored in memory during forward propagation in order to calculate the gradients of the pretrained weights ($\nabla \mathbf{W}$) in backward propagation, as shown in Eq. 3. Although LoRA does not require the computation of gradients for the pretrained weights, the input activations ($\mathbf{X}_{in}$) must still be stored in memory to calculate the gradients for the LoRA $\mathbf{A}$ layer ($\nabla \mathbf{A}$), as indicated in Eq. 6. Additionally, the output activations of the LoRA $\mathbf{A}$ layer ($\mathbf{X}_{mid}$) must be stored in memory to calculate the gradients for the LoRA $\mathbf{B}$ layer ($\nabla \mathbf{B}$) following Eq. 6. This issue with activation memory becomes more critical when training on long sequence data or increasing batch size to improve training throughput (Chen et al., 2023; Korthikanti et al., 2023; Woo et al., 2024).

## 3 METHODOLOGY

### 3.1 PaCA: PARTIAL CONNECTION ADAPTATION

Motivated by the observation that the newly introduced adapter layers lead to training inefficiencies, we propose **Pa**rtial **C**onnection **A**daptation (**PaCA**). PaCA fine-tunes randomly selected partial connections within the pretrained weights rather than introducing new adapter layers, as depicted in Fig. 1. More specifically, PaCA employs the training algorithm below:

$$\text{Forward:} \quad \mathbf{X}_{out} = \mathbf{W}\mathbf{X}_{in} \tag{7}$$

$$\text{Backward:} \quad \nabla \mathbf{X}_{in} = \mathbf{W}^T \nabla \mathbf{X}_{out} \tag{8}$$

$$\nabla \mathbf{P} = \nabla \mathbf{X}_{out}\, {}^{p}\mathbf{X}_{in}^T \tag{9}$$

where $\mathbf{P} \in \mathbb{R}^{d_{out} \times r}$ and ${}^{p}\mathbf{X}_{in} \in \mathbb{R}^r$ denote the partial connections randomly selected from the pretrained weights (i.e., $\mathbf{P} \subset \mathbf{W}$) and the corresponding partial activations selected from the input activations (i.e., ${}^{p}\mathbf{X}_{in} \subset \mathbf{X}_{in}$), respectively. $r$ represents the number of the randomly selected columns within the pretrained weights, which we refer to *rank* when PaCA is applied. The operations involving partial connections are highlighted in red.

PaCA randomly selects the partial connections to fine-tune from the pretrained weights before training and then fine-tunes only the selected connections. Since these partial connections are part of the pretrained weights, no additional computations are required in forward and backward computations (Eqs. 7-8), completely avoiding inefficient sequential processing due to the adapter layers in

LoRA. In addition, while LoRA requires both the input activations ($\mathbf{X}_{in}$) and the output activations of the LoRA $\mathbf{A}$ layer ($\mathbf{X}_{mid}$) to calculate gradients for the low-rank adapter matrices (Eq. 6), PaCA only needs to store the partial activations ($^{p}\mathbf{X}_{in}$) to calculate the gradients of the partial connections ($\nabla \mathbf{P}$), significantly reducing the amount of activation to be temporarily stored in memory.

We calculated the FLOPs and measured the training time required for fine-tuning the LLaMA3-8B model using PaCA to demonstrate its effectiveness (see Table 8 in Appendix C for experiment details), and the results are summarized in Fig. 2. Experimental results indicate that PaCA provides a 19% reduction in total training time compared to LoRA, by reducing forward propagation time by 18% and backward propagation time by 20%, achieved through avoiding additional sequential processing. One interesting observation is that while the FLOPs required for forward and backward propagation in PaCA are nearly identical, the actual runtime for backward propagation is 17% longer than forward propagation. We hypothesize that, even though the computation of weight gradients for partial connections (Eq. 9) is significantly smaller than that for the pretrained weights, it occurs sequentially with the input gradient computation (Eq. 8) during backward propagation. This sequential processing introduces additional latency compared to forward propagation, which only involves the computation of output activations (Eq. 1). It should be noted that this latency overhead is not a specific overhead introduced by PaCA, but rather an inherent issue in all backpropagation-based training algorithms including Full-FT and prior PEFT algorithms, which must compute both input gradients and weight gradients.

Intuitively, training only a subset of connections can be interpreted as learning within a subspace composed of the selected connections. Prior studies revealed that overparameterized models can be efficiently trained even when weights are projected onto a small subspace (Li et al., 2018; Aghajanyan et al., 2021). Similarly, LoRA (Hu et al., 2022) was suggested based on the assumption that weight updates can be projected onto a small low-rank subspace. Inspired by these observations, we hypothesized that weight updates could also be projected onto a small subspace composed of a subset of weight columns. In other words, we assumed that the critical factor is learning within a small subspace, not the method of selecting the subspace itself. Here we prove that training only a subset of connections is sufficient to ensure the convergence of loss in neural networks, as demonstrated in Section 3.2.

## 3.2 Convergence Analysis of PaCA

In Section 3.1, we proposed PaCA and demonstrated its effectiveness. Now we theoretically prove that PaCA converges for general neural networks. We first define the input at the $k$-th iteration as $\mathbf{X}^k$ and the full set of weights as $\mathbf{W}^k = [\mathbf{W}_1^k, \mathbf{W}_2^k, \ldots, \mathbf{W}_n^k]$, where $n$ denotes the number of layers. The loss of the model is defined as $f(\mathbf{X}^k, \mathbf{W}^k)$. The weight of the $l$-th layer $\mathbf{W}_l^k$ can be represented as a collection of column vectors (i.e., $\mathbf{W}_l^k = [_1\mathbf{w}_l^k, _2\mathbf{w}_l^k, \ldots, _{d_l}\mathbf{w}_l^k]$). In PaCA, we only fine-tune randomly selected columns $\mathbf{P}_l^k = [_{i_1}\mathbf{w}_l^k, _{i_2}\mathbf{w}_l^k, \ldots, _{i_r}\mathbf{w}_l^k]$ where $i_1, \ldots, i_r$ denote the selected column indices for PaCA. The weights are then updated as follows:

$$\text{Full-FT:} \quad \mathbf{W}_l^{k+1} = \mathbf{W}_l^k - \eta\nabla\mathbf{W}_l^k = \mathbf{W}_l^k - \eta[\nabla_1\mathbf{w}_l^k, \nabla_2\mathbf{w}_l^k, \ldots, \nabla_{d_l}\mathbf{w}_l^k] \quad (10)$$

$$\text{PaCA:} \quad \mathbf{W}_l^{k+1} = \mathbf{W}_l^k - \eta\Delta\mathbf{W}_l^k = \mathbf{W}_l^k - \eta[\mathbf{0}, \nabla_{i_1}\mathbf{w}_l^k, \ldots, \nabla_{i_r}\mathbf{w}_l^k, \ldots \mathbf{0}] \quad (11)$$

where $\eta$ denotes learning rate and $\Delta\mathbf{W}_l^k$ denotes weight updates. In this scenario, we define the full set of partial connections within the model as $\mathbf{P}^k = [\mathbf{P}_1^k, \mathbf{P}_2^k, \ldots, \mathbf{P}_n^k]$. Then, PaCA satisfies the following theorem:

**Theorem 1.** *If the gradient of the loss function $f(\mathbf{W}, \mathbf{X})$ is Lipschitz continuous and the only partial connections are updated, then*

$$f(\mathbf{W}^{k+1}, \mathbf{X}^{k+1}) \leq f(\mathbf{W}^k, \mathbf{X}^k) - \eta(1 - \frac{\eta L}{2})||\nabla\mathbf{P}^k||^2$$

We prove Theorem 1 by applying Eq. 11 to the quadratic upper bound using Lipschitz continuity condition (i.e., $f(\mathbf{W}^{k+1}, \mathbf{X}^{k+1}) \leq f(\mathbf{W}^k, \mathbf{X}^k) + \nabla_{\mathbf{W}^k}f(\mathbf{W}^k, \mathbf{X}^k)(\mathbf{W}^{k+1} - \mathbf{W}^k)^T + L/2||\mathbf{W}^{k+1} - \mathbf{W}^k||^2$) where $L$ denotes the Lipschitz constant. The detailed proof can be found in Appendix A.

Theorem 1 implies that as long as the learning rate $\eta$ is chosen to satisfy the condition $0 < \eta < 2/L$, the loss function $f(\mathbf{W}, \mathbf{X})$ will decrease after each iteration, ensuring convergence of the neural network.

# 4 EXPERIMENTS

To verify the effectiveness of PaCA, here we evaluate its performance in various fine-tuning scenarios. Section 4.1 first compares the accuracy and performance of PaCA with other PEFT algorithms, such as LoRA (Hu et al., 2022), DoRA (Liu et al., 2024), and MosLoRA (Wu et al., 2024), when fine-tuning the LLaMA2-7B/13B (Touvron et al., 2023) and LLaMA3-8B (Dubey et al., 2024) models on the MMLU dataset (Hendrycks et al., 2021). In Section 4.2, we observe the instruction-following ability on the MT-Bench dataset (Zheng et al., 2023) after fine-tuning the LLaMA3-8B model with PaCA and the LoRA family on the Oasst1 dataset (Köpf et al., 2023). In Section 4.3, we compare the performance and score of our quantized PaCA (QPaCA) with QLoRA (Dettmers et al., 2023) on the MT-Bench (Zheng et al., 2023) dataset while fine-tuning the LLaMA3.1-70B (Dubey et al., 2024) model on the Oasst1 dataset. Section 4.4 analyzes the ability of PaCA and the LoRA family to handle long sequence data and the training throughput when increasing the batch size, using both a single NVIDIA A100 (Choquette et al., 2021) and Intel Gaudi2 HPU (Intel Corporation, 2023). In addition, we tested PaCA on different model architectures such as the vision transformer (ViT (Dosovitskiy et al., 2021)) and convolutional neural network (EfficientNet-V2 (Tan & Le, 2021)) for demonstrating generalizability of PaCA in Appendix B

## 4.1 FINE-TUNING FOR SPECIFIC TASKS

We first compared PaCA against LoRA, DoRA, and MosLoRA using the MMLU dataset, which consists of 57 tasks designed to assess the ability of a model to understand and reason across a wide range of academic subjects (Hendrycks et al., 2021). The evaluation was conducted on the LLaMA2-7B/13B and LLaMA3-8B models, with the rank of the prior PEFT methods set to 8. We employ PaCA with a rank of 8 and 16, each representing the case where the rank is equal to that of prior PEFT methods and where the number of trainable parameters is identical. Aside from adjusting the learning rate for each PEFT model, all other experimental settings remained identical, as detailed in Table 9 in Appendix C. All experiments were conducted on a single NVIDIA A100 GPU.

Table 1: Comparisons of memory usage (Mem), training time (Time), and 5-shot accuracy on MMLU dataset when fine-tuning LLaMA2-7B/13B and LLaMA3-8B models using various PEFT algorithms. Param indicates the number of trainable parameters.

| Model | Method | Rank | Param | Mem | Time | Accuracy (%) | | | | |
|---|---|---|---|---|---|---|---|---|---|---|
| | | | | | | Hums. | STEM | Social. | Other | Avg. |
| LLaMA2-7B | No tuning | - | - | - | - | 44.0 | 37.0 | 51.5 | 53.1 | 45.9 |
| | LoRA | 8 | 20M | 23G | 4.1h | 48.5 | 41.2 | 57.3 | 56.5 | 50.6 |
| | DoRA | 8 | 21M | 29G | 8.7h | 48.7 | 42.3 | 58.3 | 57.6 | **51.3** |
| | MosLoRA | 8 | 20M | 23G | 4.3h | 46.6 | 42.2 | 60.8 | 57.4 | 51.1 |
| | PaCA (Ours) | 8 | 11M | **20G** | **3.2h** | 46.8 | 41.1 | 58.4 | 57.3 | 50.4 |
| | | 16 | 22M | **20G** | **3.2h** | 48.7 | 41.7 | 58.7 | 57.6 | 51.2 |
| LLaMA2-13B | No tuning | - | - | - | - | 53.1 | 44.2 | 62.8 | 60.8 | 54.9 |
| | LoRA | 8 | 31M | 40G | 6.3h | 53.9 | 46.2 | 66.8 | 62.9 | 57.0 |
| | DoRA | 8 | 33M | 49G | 14.7h | 55.6 | 46.8 | 66.7 | 64.8 | **58.1** |
| | MosLoRA | 8 | 31M | 40G | 6.5h | 56.5 | 47.3 | 66.1 | 62.8 | 57.9 |
| | PaCA (Ours) | 8 | 17M | **35G** | **5.2h** | 52.7 | 46.2 | 67.1 | 63.4 | 56.8 |
| | | 16 | 34M | **35G** | **5.2h** | 56.0 | 46.7 | 66.3 | 64.0 | 58.0 |
| LLaMA3-8B | No tuning | - | - | - | - | 59.3 | 55.3 | 75.7 | 72.7 | 64.9 |
| | LoRA | 8 | 21M | 27G | 4.4h | 59.4 | 56.3 | 75.4 | 71.9 | 65.0 |
| | DoRA | 8 | 22M | 33G | 9.4h | 59.4 | 56.3 | 75.7 | 72.2 | 65.2 |
| | MosLoRA | 8 | 21M | 27G | 4.6h | 59.8 | 55.9 | 75.7 | 72.0 | 65.1 |
| | PaCA (Ours) | 8 | 11M | **23G** | **3.5h** | 59.7 | 55.7 | 76.0 | 72.3 | 65.2 |
| | | 16 | 22M | **23G** | **3.5h** | 60.2 | 55.9 | 75.8 | 72.6 | **65.4** |

The experimental results in Table 1 demonstrate that PaCA significantly reduces both memory usage and training time across all models, while maintaining accuracy comparable to the other PEFT algorithms. For the LLaMA2-7B model, PaCA achieves accuracy similar to LoRA when the rank is set to 8, despite using only half the number of trainable parameters, reducing memory usage by 13% and training time by 26% simultaneously. In this configuration, the accuracy of PaCA drops by up to 0.9% compared to LoRA variants such as DoRA and MosLoRA. However, when the rank of PaCA is increased to 16, matching the number of trainable parameters with DoRA and MosLoRA, PaCA achieves almost identical accuracy to DoRA and MosLoRA while still offering considerable reductions in memory usage and training time. Specifically, PaCA reduces memory usage by 31% and training time by 63% compared to DoRA, while offering a 13% reduction in memory usage and a 26% reduction in training time compared to MosLoRA.

A similar trend is observed in both LLaMA2-13B and LLaMA3-8B, where PaCA continues to show substantial reductions in memory usage and training time. On LLaMA2-13B, PaCA achieves comparable accuracy to the LoRA variants while reducing memory usage by 13%, 29%, and 13%, and training time by 17%, 64%, and 20%, compared to LoRA, DoRA, and MosLoRA, respectively. In LLaMA3-8B, PaCA consumes the least memory and training time among LoRA and its variants, while achieving the highest accuracy. In summary, PaCA successfully improves training speed by eliminating unnecessary sequential processes and reduces memory usage by storing only partial activations, while maintaining comparable accuracy in fine-tuning scenarios on specific tasks.

## 4.2 INSTRUCTION TUNING

We next evaluate PaCA on the MT-Bench dataset, which consists of 80 queries designed to measure the instruction-following capabilities of a model across multiple tasks, providing a detailed assessment of its performance in real-world scenarios (Zheng et al., 2023). Specifically, we fine-tuned LLaMA3-8B using a single NVIDIA A100 GPU on the Oasst1 dataset, which is an instruction-following dataset, and then evaluated the score on the MT-Bench dataset using GPT4o-mini as the judge. The detailed setup can be found in Table 10 in Appendix C.

Table 2: Comparisons of memory usage (Mem), training time (Time), and score on MT-Bench dataset when fine-tuning LLaMA3-8B on Oasst1 dataset using various PEFT algorithms.

| Method | Rank | Mem | Time | Human. | STEM | Role. | Extract. | Writing | Reason. | Coding | Math | Avg. |
|---|---|---|---|---|---|---|---|---|---|---|---|---|
| No tuning | - | - | - | 6.25 | 5.70 | 5.45 | 4.85 | 5.20 | 4.40 | 3.20 | 1.95 | 4.62 |
| LoRA | 64 | 56G | 26m | 7.00 | 6.40 | 5.70 | 5.80 | 5.30 | 4.55 | 3.25 | 2.95 | 5.12 |
| DoRA | 64 | 65G | 50m | 6.95 | 6.00 | 5.90 | 5.80 | 6.20 | 4.50 | 3.50 | 3.40 | **5.28** |
| MosLoRA | 64 | 56G | 27m | 6.90 | 6.50 | 5.80 | 5.70 | 5.55 | 4.90 | 3.10 | 2.75 | 5.15 |
| PaCA (Ours) | 64 | **47G** | **21m** | 6.50 | 6.30 | 5.90 | 5.95 | 5.65 | 4.80 | 3.70 | 3.05 | 5.23 |
| | 128 | 51G | **21m** | 6.80 | 6.15 | 6.05 | 5.95 | 5.85 | 4.65 | 3.45 | 3.15 | 5.26 |

Table 2 confirms that PaCA significantly reduces memory usage and training time compared to other PEFT methods while maintaining comparable scores, consistent with the results observed when fine-tuning it on the MMLU dataset. Specifically, our PaCA outperforms LoRA and MosLoRA with 16% less memory usage and 19% shorter training time. Furthermore, PaCA reduces memory usage by 28% and training time by 58% compared to DoRA, while achieving comparable scores. One interesting observation is that the memory usage of PaCA increases by approximately 4GB when the rank is raised from 64 to 128, whereas the memory usage remains almost unchanged when increasing the rank from 8 to 16 in Section 4.1. This is because a higher rank requires more optimizer state memory and activation memory for fine-tuning the partial connections.

## 4.3 QPACA: ENHANCEMENTS TO QLORA

While PEFT significantly reduces the memory required for gradients and optimizer states, the model weights must be loaded onto the GPU, which consumes a significant amount of memory, especially when training large models. For example, loading the weights of LLaMA3.1-70B requires 140GB of memory, making it impossible to fine-tune using a single NVIDIA A100 GPU. To address this issue, QLoRA (Dettmers et al., 2023) quantizes the pretrained weights to 4 bits to further reduce memory usage and trains only the 16-bit adapter layers, enabling the fine-tuning of LLaMA3.1-70B on a single NVIDIA A100 GPU. This approach can be extended to PaCA by quantizing the unselected

Table 3: Comparisons of memory usage (Mem), training time (Time), and score on MT-Bench dataset when fine-tuning LLaMA3-8B and LLaMA3.1-70B on Oasst1 dataset using QLoRA and QPaCA. No tuning and Quantized in the table refer to the models in 16-bit precision without quantization and with 4-bit NormalFloat Quantization (NF), respectively, without fine-tuning.

| Model | Method | Mem | Time | Hums. | STEM | Role. | Extract. | Writing | Reason. | Coding | Math | Avg. |
|-------|--------|-----|------|-------|------|-------|----------|---------|---------|--------|------|------|
| **LLaMA3 -8B** | No tuning | - | - | 6.25 | 5.70 | 5.45 | 4.85 | 5.20 | 4.40 | 3.20 | 1.95 | 4.62 |
| | Quantized | - | - | 4.70 | 4.80 | 4.60 | 5.00 | 4.65 | 4.05 | 3.60 | 1.85 | 4.16 |
| | QLoRA | 18G | 42m | 6.85 | 5.75 | 5.85 | 6.00 | 5.15 | 4.70 | 3.35 | 2.35 | 5.00 |
| | QPaCA | **16G** | **37m** | 6.85 | 5.95 | 5.65 | 5.60 | 5.15 | 4.05 | 3.65 | 3.25 | **5.02** |
| **LLaMA3.1 -70B** | Quantized | - | - | 7.40 | 7.05 | 5.85 | 6.50 | 6.85 | 5.30 | 4.60 | 3.80 | 5.92 |
| | QLoRA | 80G | 5.1h | 7.40 | 6.85 | 6.55 | 7.20 | 6.55 | 5.65 | 4.75 | 3.80 | **6.09** |
| | QPaCA | **69G** | **4.7h** | 7.70 | 7.40 | 6.40 | 6.80 | 6.50 | 5.40 | 4.75 | 3.70 | 6.08 |

connections within the pretrained weights to 4 bits, while fine-tuning only the 16-bit randomly selected partial connections. We named this algorithm Quantized Partial Connection Adaptation (QPaCA) and compared it with QLoRA when fine-tuning LLaMA3-8B and LLaMA3.1-70B on the Oasst1 dataset using a single NVIDIA A100 GPU. Following Section 4.2, we evaluated the score on the MT-Bench dataset, using GPT4o-mini as the judge. Further details can be found in Table 11 in Appendix C.

Experimental results demonstrate that QPaCA reduces both memory usage and training time compared to QLoRA, as displayed in Table 3. Specifically, on the LLaMA3-8B model, QPaCA not only achieved higher scores than the model quantized in the NF4 format, but also outperformed the 16-bit baseline, similar to QLoRA. Furthermore, QLoRA achieved an 11% reduction in memory usage and a 12% reduction in training time compared to QPaCA.

In addition, even on a larger scale model, LLaMA3.1-70B, QPaCA successfully reduces memory usage by 14% and training time by 8% with almost no drop in score compared to QLoRA and higher scores than the NF4 quantized model without fine-tuning on the MT-Bench dataset. This training time reduction is relatively smaller than when comparing PaCA with LoRA in previous sections, and this is due to the time overheads of additional quantization and dequantization processes, which cannot be reduced by training only partial connections, unlike the forward and backward propagations.

## 4.4 USABILITY OF PACA

Table 4: Max sequence length for fine-tuning LLaMA3-8B using vaious PEFT algorithms on a single NVIDIA A100 GPU.

| Method | LoRA | DoRA | MosLoRA | PaCA (Ours) |
|--------|------|------|---------|-------------|
| **Max Length** | 8.0K | 4.7K | 8.0K | **9.8K** |

In this section, we evaluate the usability of PaCA by measuring its training performance in different scenarios. We first increase the sequence length of the data while fine-tuning the LLaMA3-8B model with each PEFT method until an out-of-memory (OOM) error occurs, and the maximum sequence length is displayed in Table 4. For a fair comparison, all other conditions, such as batch size and rank, were kept constant, except for the sequence length, as detailed in Table 12 in Appendix C. We found that PaCA increased the maximum sequence length by 23%, 108%, and 23% compared to LoRA, DoRA, and MosLoRA, respectively, by storing only partial activations instead of all input activations.

Next, we evaluate the training throughput improvements achieved by PaCA compared to LoRA and its variants as the batch size increases when fine-tuning LLaMA3-8B using a single NVIDIA A100 GPU and Intel Gaudi2 HPU. Specifically, we kept all configurations identical except for the batch size as presented in Table 13 in Appendix C, and measured the throughput as the batch size increased for each PEFT method until an OOM error occurred. As shown in Fig. 3, PaCA demonstrated the ability to increase the batch size by 33% on the NVIDIA A100 GPU and 21% on the Intel Gaudi2 HPU compared to LoRA and its variants, primarily due to its reduction of activation memory. This

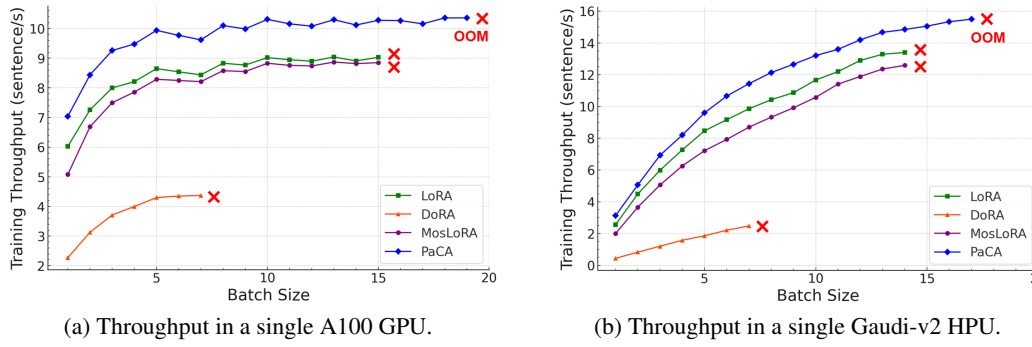

(a) Throughput in a single A100 GPU.  (b) Throughput in a single Gaudi-v2 HPU.

Figure 3: Training throughput (sentences/s) on a single NVIDIA A100 GPU and INTEL Gaudi2 HPU when fine-tuning LLaMA3-8B with a sequence length of 512.

reduction allows PaCA to handle larger batch sizes, which directly leads to better resource utilization and improves scalability. In addition, at the same batch size, PaCA consistently achieved higher training throughput compared to LoRA and its variants, as PaCA eliminates inefficient sequential processing introduced by adapter layers, allowing for higher hardware utilization. Consequently, PaCA outperformed LoRA, achieving a throughput of 10.36 sentences/s on A100 GPU and 15.5 sentences/s on Gaudi2 HPU, representing a 16% improvement for both GPUs.

## 5 EFFECT OF SELECTION STRATEGY

In this section, we explore alternative strategies for selecting connections in PaCA and evaluate their effectiveness beyond the random selection approach. We tested two selection schemes that consider the importance of each column. A weight-based strategy selects the columns with the highest $L_2$-Norm from the initial pretrained weights, whereas a gradient-based strategy accumulates gradients during the first 100 iterations without updating weights (i.e., $G_i = \sum_t \|g_i^t\|^2$, where $i$ is the number of layers and $t$ is the accumulation step) and selects columns with the largest accumulated gradients. Experimental results are displayed in the table below.

Table 5: Test score on MT-Bench dataset when fine-tuning LLaMA3-8B with PaCA using various connection selecting strategy on Oasst1 dataset.

| Method | Human. | STEM | Role. | Extract. | Writing | Reason. | Coding | Math | Avg. |
|---|---|---|---|---|---|---|---|---|---|
| No tuning | 6.25 | 5.70 | 5.45 | 4.85 | 5.20 | 4.40 | 3.20 | 1.95 | 4.62 |
| Random (Seed #1) | 6.50 | 6.30 | 5.90 | 5.95 | 5.65 | 4.8 | 3.7 | 3.05 | 5.23 |
| Random (Seed #2) | 6.50 | 6.00 | 6.30 | 5.90 | 5.70 | 4.90 | 3.80 | 3.00 | **5.26** |
| Weight-based | 7.00 | 5.70 | 6.05 | 5.80 | 5.70 | 4.55 | 3.90 | 2.70 | 5.18 |
| Gradient-based | 6.95 | 6.40 | 6.25 | 5.35 | 5.95 | 4.55 | 3.80 | 2.70 | 5.24 |

Table 5 demonstrates that random selection achieves similar performance to importance-based selection schemes. In other words, the choice of selection strategy does not noticeably affect fine-tuning accuracy. Therefore, we chose to select connections randomly in PaCA, as this strategy eliminates the need for complex processes to measure the importance of connections, thereby minimizing training time or memory overhead without performance degradation.

## 6 RELATED WORK

**Parameter-efficient fine-tuning (PEFT)** Fine-tuning LLMs requires significant memory resources to store parameter gradients and optimizer states. PEFT algorithms address this challenge by introducing adapter layers with far fewer parameters than the pretrained models, significantly reducing the memory required for parameter gradients and optimizer states by fine-tuning only the

adapter layers. PEFT methods can generally be categorized into three groups: *Adapter-based methods* (Li & Liang, 2021; Houlsby et al., 2019; He et al., 2022), *Prompt-based methods* (Lester et al., 2021; Razdaibiedina et al., 2023; Wang et al., 2023; Zhang et al., 2024; Gao et al., 2023), and *LoRA and its variants* (Hu et al., 2022; Kopiczko et al., 2024; Liu et al., 2024; Wu et al., 2024). *Adapter-based methods* introduce new trainable adapter weights to the pretrained models. For example, Houlsby et al. (2019) adds adapter layers as linear modules in series with the existing model, while He et al. (2022) inserts adapter modules in parallel with the pretrained model. Secondly, *Prompt-based methods* inject new trainable prompt vectors into the model. Specifically, LLaMA-Adapter (Zhang et al., 2024; Gao et al., 2023) introduces prompts into the upper layers of the transformer, enabling the model to incorporate diverse knowledge. Although these approaches enable efficient fine-tuning with smaller trainable parameters, they introduce latency overhead during inference due to the sequential processing of the adapter layers and the pretrained model.

The third category of PEFT methods is *LoRA and its variants*. LoRA (Hu et al., 2022) introduces low-rank matrices as adapters to approximate weight gradients during fine-tuning, then merges these low-rank matrices with the pretrained weights, effectively eliminating inference overhead. VeRA (Kopiczko et al., 2024) takes this approach further by freezing the low-rank matrices and sharing them across layers, while only learning the scaling vectors for each layer, which significantly reduces the number of trainable parameters. DoRA (Liu et al., 2024) improves upon LoRA by considering both the magnitude and direction of gradients through weight decomposition, leading to higher accuracy compared to LoRA. MosLoRA (Wu et al., 2024) enhances LoRA by introducing a learnable mixer between the two low-rank matrices, improving its capabilities. SHiRA (Bhardwaj et al., 2024) fine-tunes a sparse 1–2% subset of the pretrained weights, thereby enabling rapid adapter switching during inference in mobile environments. Even though *LoRA and its variants* can remove latency overhead by merging the adapter weights with the pretrained weights during inference, latency overhead persists during fine-tuning, as merging the weights is not feasible at this stage.

**PEFT with Quantization**  Quantization (Dettmers et al., 2022; Frantar et al., 2022; Lin et al., 2024; Frantar et al., 2023) is a technique that reduces memory usage and computational complexity by representing weights or activations in low precision. This method can also be combined with PEFT to reduce memory usage during fine-tuning (Kwon et al., 2022; Dettmers et al., 2023; Xu et al., 2024). For instance, QLoRA (Dettmers et al., 2023) compresses pretrained weights to 4 bits and trains only the low-rank adapter matrices represented in 16 bits, significantly reducing the memory required to load the model. Additionally, QA-LoRA (Xu et al., 2024) integrates the low-rank adapter matrices with the zero point in quantization, enabling the direct generation of a 4-bit quantized model after fine-tuning. While those quantized-PEFT approaches reduce memory usage for fine-tuning, the sequential processes introduced by the adapter layers still cause training time overhead.

## 7    CONCLUSION

In this work, we propose PaCA, a memory-efficient PEFT algorithm that fine-tunes randomly selected partial connections within the pretrained weights without employing additional adapter layers. By removing the sequential processing overhead associated with the adapters in prior PEFT schemes, PaCA significantly improves hardware utilization and training speed. In addition, PaCA reduces activation memory by only storing partial activations instead of all input activations. We theoretically prove that PaCA can successfully converge in general deep neural networks. Moreover, in experiments, PaCA consistently outperforms LoRA and its variants in training performance while maintaining comparable accuracy across various fine-tuning scenarios. We also show that PaCA can be applied simultaneously with quantization. Finally, we demonstrate the effectiveness of PaCA in scenarios involving long sequence data or when maximizing throughput in resource-constrained environments. For future work, we aim to develop methods for identifying optimal partial connections in PaCA, rather than relying on random selection, to further enhance fine-tuning accuracy.

REPRODUCIBILITY

We introduce PaCA and provide a detailed explanation of its concept and potential in Section 3.1, and prove its theoretical convergence in Section 3.2. In addition, the setup and hyperparameters are thoroughly explained in Section 4 and Appendix C. Furthermore, we have implemented PaCA using PyTorch (Paszke et al., 2019), a widely used deep learning framework, and integrated it into the PEFT library in Huggingface (Wolf et al., 2019) to ensure easy reproducibility.

ACKNOWLEDGMENT

This work was supported by the National Research Foundation of Korea (Grant NRF-2022R1C1C1006880), the Institute of Information & Communications Technology Planning & Evaluation (Grant IITP-2023-RS-2023-00256081 and Grant RS-2024-00347394), and the NAVER-Intel Co-Lab.

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

APPENDICES

## A   PROOF FOR CONVERGENCE OF PACA

**Theorem 1.** *If the gradient of the loss function $f(\boldsymbol{W}, \boldsymbol{X})$ is Lipschitz continuous and the only partial connections are updated, then*

$$f(\boldsymbol{W}^{k+1}, \boldsymbol{X}^{k+1}) \leq f(\boldsymbol{W}^k, \boldsymbol{X}^k) - \eta(1 - \frac{\eta L}{2})||\nabla \boldsymbol{P}^k||^2$$

*Proof.* As the gradient of the loss function $f(\mathbf{W}, \mathbf{X})$ is Lipschitz continuous, we obtain

$$f(\mathbf{W}^{k+1}, \mathbf{X}^{k+1}) \leq f(\mathbf{W}^k, \mathbf{X}^k) + \nabla_{\mathbf{W}^k} f(\mathbf{W}^k, \mathbf{X}^k)^T (\mathbf{W}^{k+1} - \mathbf{W}^k) + \frac{L}{2}||\mathbf{W}^{k+1} - \mathbf{W}^k||^2$$

By substituting Eq. 11 which represents partial connection updates, we obtain

$$
\begin{aligned}
f(\mathbf{W}^{k+1}, \mathbf{X}^{k+1}) &\leq f(\mathbf{W}^k, \mathbf{X}^k) + \nabla_{\mathbf{W}^k} f(\mathbf{W}^k, \mathbf{X}^k)(\mathbf{W}^{k+1} - \mathbf{W}^k)^T + \frac{L}{2}||\mathbf{W}^{k+1} - \mathbf{W}^k||^2 \\
&= f(\mathbf{W}^k, \mathbf{X}^k) + \nabla_{\mathbf{W}^k} f(\mathbf{W}^k, \mathbf{X}^k)(-\eta \Delta \mathbf{W}^k)^T + \frac{L}{2}|| - \eta \Delta \mathbf{W}^k||^2 \\
&= f(\mathbf{W}^k, \mathbf{X}^k) - \eta(\nabla_{\mathbf{W}^k} f(\mathbf{W}^k, \mathbf{X}^k) - \frac{\eta L}{2} \Delta \mathbf{W}^k)(\Delta \mathbf{W}^k)^T \\
&= f(\mathbf{W}^k, \mathbf{X}^k) - \sum_{l=1}^{n} \eta(\nabla_{\mathbf{W}_l^k} f(\mathbf{W}^k, \mathbf{X}^k) - \frac{\eta L}{2} \Delta \mathbf{W}_l^k)(\Delta \mathbf{W}_l^k)^T \\
&= f(\mathbf{W}^k, \mathbf{X}^k) - \sum_{l=1}^{n} \eta(\nabla_{\mathbf{W}_l^k} f(\mathbf{W}^k, \mathbf{X}^k) - \frac{\eta L}{2} \Delta \mathbf{W}_l^k)(\Delta \mathbf{W}_l^k)^T
\end{aligned}
$$

Also, $\nabla_{\mathbf{W}_l^k} f(\mathbf{W}^k, \mathbf{X}^k)$ and $\Delta \mathbf{W}_l^k$ can be expressed as

$$\nabla_{\mathbf{W}_l^k} f(\mathbf{W}^k, \mathbf{X}^k) = \left[ {}_m \nabla w_l^k \right]_{m=1}^{d_l}$$

$$\Delta \mathbf{W}_l^k = \left[ {}_m \nabla w_l^k \text{ if } m \in I = \{i_1, i_2, \dots, i_r\}, \text{ else } \mathbf{0} \right]_{m=1}^{d_l}$$

where $I$ represents the set of indices corresponding to the selected columns. By applying $\nabla_{\mathbf{W}_l^k} f(\mathbf{W}^k, \mathbf{X}^k)$ and $\Delta \mathbf{W}_l^k$ above, we obtain

$$
\begin{aligned}
f(\mathbf{W}^{k+1}, \mathbf{X}^{k+1}) &\leq f(\mathbf{W}^k, \mathbf{X}^k) - \sum_{l=1}^{n} \eta(\nabla_{\mathbf{W}_l^k} f(\mathbf{W}^k, \mathbf{X}^k) - \frac{\eta L}{2} \Delta \mathbf{W}_l^k)(\Delta \mathbf{W}_l^k)^T \\
&= f(\mathbf{W}^k, \mathbf{X}^k) - \sum_{l=1}^{n} \eta \left[ (1 - \frac{\eta L}{2}) {}_m \nabla w_l^k \text{ if } m \in I, \text{ else } {}_m \nabla w_l^k \right]_{m=1}^{d_l} (\Delta \mathbf{W}_l^k)^T \\
&= f(\mathbf{W}^k, \mathbf{X}^k) - \sum_{l=1}^{n} \sum_{m \in I} \eta(1 - \frac{\eta L}{2})||{}_m \nabla w_l^k||^2 \\
&= f(\mathbf{W}^k, \mathbf{X}^k) - \sum_{l=1}^{n} \eta(1 - \frac{\eta L}{2})||\nabla \mathbf{P}_l^k||^2 = f(\mathbf{W}^k, \mathbf{X}^k) - \eta(1 - \frac{\eta L}{2})||\nabla \mathbf{P}^k||^2
\end{aligned}
$$

$\square$

We assumed the Lipschitz continuity of gradients to theoretically prove the convergence of PaCA. However, we acknowledge the inherent limitations of the Lipschitz continuity assumption. In practice, this assumption may not hold for certain neural networks, particularly in scenarios where gradient magnitudes vary significantly due to sharp activation functions, high model complexity, or specific architectural designs. It is well known that it is very challenging to theoretically analyze the convergence of general deep neural networks. Therefore, prior studies (Belilovsky et al., 2020; Chen et al., 2021; Liu et al., 2022; Woo & Jeon, 2023) first proved the convergence of the proposed algorithm under weak constraints, such as the Lipschitz continuity of gradients, and then validated convergence empirically in real-world scenarios.

Following a similar approach, we assumed the Lipschitz continuity of gradients to theoretically prove the convergence of PaCA. Then, we experimentally demonstrated that PaCA successfully trains real-world large-scale neural networks such as LLaMA Models, where the Lipschitz continuity assumption may not strictly hold, as shown in Tables 1-3 in Section 4.

## B  APPLICABILITY OF PACA TO OTHER ARCHITECTURES AND TASKS

In this section, we fine-tune ViT-B/16 (Dosovitskiy et al., 2021) and EfficientNetV2-L (Tan & Le, 2021) using various datasets such as CIFAR-10 (Krizhevsky & Hinton, 2009), CIFAR-100 (Krizhevsky & Hinton, 2009), Oxford-IIIT Pets (Parkhi et al., 2012), and Oxford-Flowers 102 (Nilsback & Zisserman, 2008) to evaluate the generalizability of PaCA.

Table 6: Comparisons of memory usage (Mem), training time (Time), and accuracy when fine-tuning ViT-B/16 on CIFAR-10, CIFAR-100, Oxford-III Pets, and Oxford-Flowers 102.

| Method | Mem | Time | Accuracy (%) | | | | |
|---|---|---|---|---|---|---|---|
| | | | CIFAR10 | CIFAR100 | IIIT Pets | Flowers102 | Avg. |
| LoRA | 11.0G | 45m | 98.9 | 92.5 | 93.6 | 99.2 | 96.1 |
| PaCA (Ours) | **6.7G** | **32m** | 98.9 | 92.8 | 93.9 | 99.1 | **96.2** |

Table 7: Comparisons of memory usage (Mem), training time (Time), and accuracy when fine-tuning EfficientNetV2-L on CIFAR-10 and CIFAR-100.

| Method | Mem | Time | Accuracy (%) | | |
|---|---|---|---|---|---|
| | | | CIFAR10 | CIFAR100 | Avg. |
| Full-FT | 18.3 GB | 70m | 98.5 | 90.1 | **94.3** |
| PaCA (Ours) | **13.2 GB** | **59m** | 98.0 | 89.3 | 93.7 |

Table 6 shows that our PaCA achieves comparable accuracy to LoRA while reducing training memory and time by 39% and 29%, respectively, on the ViT-B/16 model. Similarly, in Table 7, PaCA demonstrated its effectiveness on EfficientNetV2-L, achieving comparable accuracy while saving 28% in training memory and 16% in training time compared to full fine-tuning.

It should be noted that conventional PEFT algorithms such as LoRA face critical limitations when applied to convolutional neural networks since the additional adapters in LoRA are implemented as linear layers, which makes it impossible to directly merge them into a pretrained layer in a different type (e.g., convolutional layer) during inference. In contrast, PaCA fine-tunes a subset of the existing pretrained weights, enabling seamless applications to diverse types of layers including convolutional layers, ensuring its generalizability.

## C  EXPERIMENTAL DETAILS

Table 8: Hyperparameters used for analyzing the number of operations and the average training time per iteration, averaged over 100 iterations, for fine-tuning LLaMA3-8B.

| Hyperparameters | Full-FT | LoRA | PaCA |
|---|---|---|---|
| Training Precision | | 16 bits | |
| Rank | | 8 | |
| Batch Size per Step | | 2 | |
| Sequence Length | | 512 | |
| Target Modules | Q, K, V, O, Up, Down, Gate | | |

Table 9: Hyperparameters when fine-tuning LLaMA2-7B/13B and LLaMA3-8B using PEFT algorithms on the MMLU dataset.

| Hyperparameters | LoRA | DoRA | MosLoRA | PaCA |
|---|---|---|---|---|
| Rank | 8 | 8 | 8 | 8/ 16 |
| $\alpha$ | 32 | 32 | 32 | 32/ 64 |
| DropOut | 0.1 | 0.1 | 0.1 | - |
| LR (LLaMA2-7B) | 3e-4 | 3e-4 | 3e-4 | 3e-4/ 1e-4 |
| LR (LLaMA2-13B) | 2e-4 | 1e-4 | 1e-4 | 1e-4/ 1e-4 |
| LR (LLaMA3-8B) | 1e-5 | 5e-6 | 5e-6 | 5e-6/ 5e-6 |
| Training Precision | | 16-bit mixed precision | | |
| Optimizer | | AdamW | | |
| LR Scheduler | | cosine | | |
| Batch Size | | 8 | | |
| Gradient Accumulation Steps | | 4 | | |
| Sequence Length | | 512 | | |
| Warmup Steps | | 100 | | |
| Epochs | | 1 | | |
| Target Modules | | Q, K, V, O, Up, Down, Gate | | |

Table 10: Hyperparameters used when fine-tuning LLaMA3-8B using PEFT algorithms on the Oasst1 dataset.

| Hyperparameters | LoRA | DoRA | MosLoRA | PaCA |
|---|---|---|---|---|
| Rank | 64 | 64 | 64 | 64/ 128 |
| $\alpha$ | 1 | 1 | 1 | 1 |
| DropOut | | - | | |
| Training Precision | | 16-bit mixed precision | | |
| Optimizer | | AdamW | | |
| LR | | 5e-4, 1e-3, 5e-3 | | |
| LR Scheduler | | linear | | |
| Batch Size | | 16 | | |
| Gradient Accumulation Steps | | 4 | | |
| Sequence Length | | 768 | | |
| Warmup Ratio | | 0.1 | | |
| Epochs | | 1 | | |
| Target Modules | | Q, K, V, O, Up, Down, Gate | | |

Table 11: Hyperparameters used when fine-tuning LLaMA3.1-70B using QLoRA and QPaCA on the Oasst1 dataset.

| Hyperparameters | LLaMA-8B | LLaMA3.1-70B |
|---|---|---|
| Gradient Accumulation Steps | 4 | 2 |
| Rank | 64 | |
| $\alpha$ | 1 | |
| DropOut | - | |
| Training Precision | 16-bit mixed precision | |
| Optimizer | AdamW | |
| LR | 5e-4, 1e-3, 5e-3 | |
| LR Scheduler | linear | |
| Batch Size | 16 | |
| Sequence Length | 768 | |
| Warmup Ratio | 0.1 | |
| Epochs | 1 | |
| Target Modules | Q, K, V, O, Up, Down, Gate | |

Table 12: Hyperparameters used for verifying the maximum sequence length on a single GPU for fine-tuning LLaMA3-8B.

| Hyperparameters | Full-FT | LoRA | DoRA | MosLoRA | PaCA |
|---|---|---|---|---|---|
| Rank | 8 | | | | |
| Training Precision | 16-bit mixed precision | | | | |
| Batch Size per Step | 1 | | | | |
| Target Modules | Q, K, V, O, Up, Down, Gate | | | | |

Table 13: Hyperparameters for comparing training throughput when increasing batch size on a single GPU for fine-tuning LLaMA3-8B.

| Hyperparameters | Full-FT | LoRA | DoRA | MosLoRA | PaCA |
|---|---|---|---|---|---|
| Rank | 8 | | | | |
| Training Precision | 16-bit | | | | |
| Sequence Length | 512 | | | | |
| Target Modules | Q, K, V, O, Up, Down, Gate | | | | |

