# OpenReview forum: "PaCA: Partial Connection Adaptation for Efficient Fine-Tuning"
_ICLR.cc/2025/Conference — ICLR 2025 Poster_

### Official Review · Reviewer_rHLZ · 2024-10-21

**Soundness:** 4
**Presentation:** 4
**Contribution:** 3
**Rating:** 6
**Confidence:** 4

**Summary:**

The authors proposed a novel method for PEFT, named Partial Connection Adaptation(PaCA). PaCA fine-tunes randomly selected partial connections in pre-trained weights instead of using adapter layers like LoRA. This innovation leads to a faster training speed and reduced memory usage while maintaining almost similar accuracy. The authors presented results on fine-tuning large models such as LLaMA3, demonstrating PaCA's ability to reduce training time by 22% and memory usage by 16%.

**Strengths:**

1. PaCA could significantly reduce memory usage and training time compared to other PEFT methods like LoRA by avoiding the need for additional adapter layers.

2. PaCA performs well with large models and long sequence data, increasing the maximum sequence length and improving throughput.

3. The authors provide theoretical analysis to demonstrate that PaCA effectively converges for general neural networks.

**Weaknesses:**

1. The authors choose to randomly select partial connections in PaCA instead of using some strategic selection. How could the random selection be generalized when fine-tuned with PaCA? Could a more targeted selection improve the performance?

2. While this paper only presents results on LLaMA models, it would be beneficial to see how PaCA performs on a wider range of architectures, such as non-transformer-based models or other tasks beyond language models.

3. Does PaCA introduce any stability issues in training, particularly when fine-tuning very large models with longer sequence lengths?

**Questions:**

see weaknesses above

---

> ### Author Response · Authors · 2024-11-24
> **Dear reviewer rHLZ**
>
> We thank the reviewer for carefully reviewing our submission and providing valuable feedback. Please see below for our response to the questions and comments.
>
> **Q4.1)** How could the random selection be generalized when fine-tuned with PaCA? Could a more targeted selection improve the performance?
>
> **A4.1)**  We thank the reviewer for this insightful suggestion. In this work, we employed random selection since the process of selecting connections could introduce significant overheads in training time and memory. For example, selecting connections to fine-tune based on parameter gradient importance requires accumulating gradients over multiple data points. This additional processing step necessitates computing the gradients for the entire connections, not just for a part of connections, and storing all activations. In contrast, our PaCA selects connections randomly without assessing their importance, resulting in negligible training time and memory overhead.
>
> In Section 3.2, we theoretically demonstrated that loss of the model can converge even when partial connections are selected randomly. Nevertheless, we entirely agree with the reviewer that an alternative approach could result in better fine-tuning performance. **Therefore, to address the reviewer’s concern, we additionally compared our random selection scheme with various other selection strategies.** More specifically, we tested two selection schemes that consider the importance of each column. A weight-based strategy selects the columns with the highest L2-Norm from the initial pretrained weights, whereas a gradient-based strategy accumulates gradients during the first 100 iterations without updating weights (i.e., $G_{i} = \sum ||g_{i}^{t}||^{2}$, where 𝑖 is the number of layers and 𝑡 is the accumulation step) and selects columns with the largest accumulated gradients. Experimental results are displayed in the table below.
>
> + Table R4.1. Test score on MT-Bench dataset when fine-tuning LLaMA3-8B with PaCA using various connection selecting strategies on Oasst1 dataset.
>
> |                  | Humanities | STEM | Roleplay | Extraction | Writing | Reasoning | Coding | Math | Avg.  |
> |------------------|------------|------|----------|------------|---------|-----------|--------|------|-------|
> | **No tuning**     | 6.25       | 5.7  | 5.45     | 4.85       | 5.2     | 4.4       | 3.2    | 1.95 | 4.62  |
> | **Random (Seed #1)**       | 6.5        | 6.3  | 5.9      | 5.95       | 5.65    | 4.8       | 3.7    | 3.05 | 5.23  |
> | **Random (Seed #2)** | 6.5        | 6.0  | 6.3      | 5.9        | 5.7     | 4.9       | 3.8    | 3.0  | **5.26**  |
> | **Weight-based** | 7.0        | 5.7  | 6.05     | 5.8        | 5.7     | 4.55      | 3.9    | 2.7  | 5.18  |
> | **Gradient-based** | 6.95       | 6.4  | 6.25     | 5.35       | 5.95    | 4.55      | 3.8    | 2.7  | 5.24  |
>
> Table R4.1 demonstrates that random selection achieves similar performance to importance-based selection schemes. In other words, the choice of selection strategy does not noticeably affect fine-tuning accuracy.
> Nevertheless, as suggested by the reviewer, exploring methods to identify the most critical partial connections for fine-tuning is an interesting research topic. In future work, we will theoretically and experimentally investigate optimal strategies for selecting connections to fine-tune in PaCA. This discussion has been added to Section 5 of the revised manuscript.

---

> ### Author Response · Authors · 2024-11-24
> **Dear reviewer rHLZ**
>
> **Q4.2)** While this paper only presents results on LLaMA models, it would be beneficial to see how PaCA performs on a wider range of architectures, such as non-transformer-based models or other tasks beyond language models.**
>
> **A4.2)** We thank the reviewer for this insightful feedback. While experimental results in the initial submission successfully show the fine-tuning capability of our algorithm on LLMs, the reviewer is certainly correct that more experiments are needed to demonstrate its generalizability. In response, we additionally tested our algorithm on the vision transformer (ViT [1]) on various datasets. We also evaluated our PaCA on a convolutional neural network (EfficientNet-V2 [2]) using the CIFAR-10 and CIFAR-100 datasets [3]. Experimental results are displayed below.
>
> + Table R4.2. Test accuracy for fine-tuning ViT-B/16 on various classification datasets.
>
> |           | Mem      | Time    | CIFAR10 | CIFAR100 | IIIT Pets | Flowers102 | AVG.  |
> |-----------|----------|---------|---------|----------|-----------|------------|-------|
> | **LoRA**  | 11.0 GB | 45m     | 98.9%    | 92.5%     | 93.6%      | 99.2%       | 96.1%  |
> | **PaCA (Ours)**  | **6.7 GB**  | **32m**     | 98.9%    | 92.8%     | 93.9%      | 99.1%       | **96.2%**  |
>
> + Table R4.3. Test accuracy for fine-tuning EfficientNetV2-L on CIFAR-10 and CIFAR-100 datasets.
>
> |               | Mem      | Time   | CIFAR10 | CIFAR100 | AVG.  |
> |---------------|----------|--------|---------|----------|-------|
> | **Full-FT**   | 18.3GB   | 70m    | 98.5%    | 90.1%     | **94.3%**  |
> | **PaCA (Ours)** | **13.2GB**   | **59m**    | 98%      | 89.3%     | **93.7%**  |
>
> Table R4.2 shows that our PaCA achieves comparable accuracy to LoRA while reducing training memory and time by 39% and 29%, respectively, on the ViT-B/16 model. Similarly, in Table R4.3, PaCA demonstrated its effectiveness on EfficientNetV2-L, achieving comparable accuracy while saving 28% in training memory and 16% in training time compared to full fine-tuning.
> It should be noted that conventional PEFT algorithms such as LoRA face critical limitations when applied to convolutional neural networks since the additional adapters in LoRA are implemented as linear layers, which makes it impossible to directly merge them into a pretrained layer of a different type (e.g., convolutional layer) during inference. In contrast, PaCA fine-tunes a subset of the existing pretrained weights, enabling seamless application to diverse types of layers, including convolutional layers. We sincerely appreciate the insightful feedback from the reviewer and have highlighted this advantage of our algorithm in Appendix B of the revised manuscript, along with additional experimental results in the tables above. We will continue to conduct further experiments on a wider range of datasets and models and make sure the results are incorporated in the final version.

---

> > ### Author Response · Authors · 2024-11-29
> > **Dear reviewer rHLZ for Q4.2**
> >
> > We sincerely apologize for the confusion caused by the initial response to Q4.2 (**A4.2**), which was mistakenly uploaded as a duplicate of a previous answer. We have now uploaded the correct response, which addresses the reviewer's question regarding PaCA's performance across a broader range of architectures, by presenting fine-tuning results of Vision Transformer (ViT) on various datasets and EfficientNet-V2 on the CIFAR-10 and CIFAR-100 datasets.
> >
> > Once again, we are truly sorry for any inconvenience this may have caused.
> >
> > Best regards,
> >
> > Authors.

---

> ### Author Response · Authors · 2024-11-24
> **Dear reviewer rHLZ**
>
> **Q4.3)** Does PaCA introduce any stability issues in training, particularly when fine-tuning very large models with longer sequence lengths?
>
> **A4.3)** We thank the reviewer for raising an important issue. Please note that, in the initial submission, we performed instruction-tuning on LLaMA3.1-70B, which is the third-largest open-source model and the largest model employed in prior studies [1, 2], using the maximum sequence length (768) achievable within our limited GPU environment. The experimental results in Table 3 of the manuscript confirm that PaCA performs well even on large-scale models. In addition, to address the reviewer’s concern, we extracted the loss curve for this experiment and the results are displayed in the figure below.
>
> + [Figure R1. Learning curves when instruction-tuning LLaMA3.1-70B with QLoRA and QPaCA on the Oasst1 dataset](https://drive.google.com/file/d/160sEUIPhuMXLVK0EkaRhQWY2c7olyP8Q/view?usp=sharing).
>
> Fig. R1 confirms that QPaCA (PaCA + quantization) converges well on LLaMA3.1-70B without any stability issues, closely matching QLoRA.
>
> In addition, to address the reviewer's request to examine stability issues with long sequences, we performed instruction-tuning on the LLaMA2-7B model using the LongAlpaca dataset. To implement long-sequence scenarios within our limited GPU environment, we adopted a training cost reduction scheme from [5], which facilitates long-sequence training by dropping some of the layers during backpropagation. As a result, the sequence length was increased to 48k. Experimental results are shown in the figure below.
>
> + [Figure R2. Learning curves for instruction-tuning LLaMA2-7B with LoRA and PaCA using DropBP on the LongAlpaca dataset.](https://drive.google.com/file/d/1xeSJSuHYtQFQPxecy0UlJRl2q0oaa3zg/view?usp=sharing)
>
> Fig. R2 shows that PaCA achieves good loss convergence similar to LoRA for a long sequence of 48k.

---

> ### Author Response · Authors · 2024-11-24
> **Dear reviewer rHLZ**
>
> [1] Dettmers et al., QLoRA: Efficient Finetuning of Quantized LLMs., Neurips2023.
>
> [2] Liu et al., DoRA: Weight-Decomposed Low-Rank Adaptation., ICML 2024.
>
> [3] Woo et al., DropBP: Accelerating Fine-Tuning of Large Language Models by Dropping Backward Propagation, Neurips 2024.

---

### Official Review · Reviewer_9boY · 2024-10-28

**Soundness:** 3
**Presentation:** 3
**Contribution:** 3
**Rating:** 6
**Confidence:** 4

**Summary:**

The authors propose PaCA, a novel PEFT algorithm selecting only partial connections within pretrained weights for finetuning, that reduces memory usage and compuational cost compared to LoRA and its variants.
The paper contains both an empirical evaluation of the effectiveness of PaCA in finetuning LLMs and a theoretical prove of the loss convergence in general neural networks when using PaCA.

**Strengths:**

- The empirical evaluation shows a significant reduction in resource consumption when using PaCA, while being able to achieve similar results to LoRA and several of its variants (DoRA, MosLoRa, ...).
- The authors include an extension of their method that works with quantized pre-trained weights, which further increases the applicability of their proposed technique.
- The proof of convergence is very useful, not only for PaCA, but also for algorithms using similar partial updating strategies.

**Weaknesses:**

- Especially for on-device training of fully quantized CNNs on embedded systems/microcontrollers, updating only partial connections is already a well-known technique [1], [2].
- Both [1] and [2] use heuristics ([1] has an offline heuristic based on XAI, [2] an online heuristic based on the magnitude of structures in feature maps) to decide which subset of weights to update compared to the potentially inferior random selection approach proposed in this work.

[1] Lin, Ji, et al. "On-device training under 256kb memory." Advances in Neural Information Processing Systems (2022).

[2] Deutel, Mark, et al. "On-Device Training of Fully Quantized Deep Neural Networks on Cortex-M Microcontrollers." IEEE Transactions on Computer-Aided Design of Integrated Circuits and Systems (2024).

**Questions:**

- For what kind of operators and NN "types" does the proposed technique apply? The paper initially focuses on MLPs ("linear layers"), but this seems to be only exemplary, as the evaluation is then performed for LLMs. Does the proposed technique also apply to e.g. CNNs ("convolutional layers")?
- Figure 1. right graph: are these real measured numbers or is the graph just symbolic? If they are real numbers, their order of magnitude should be shown on the y-axis, as in Figure 2, otherwise I do not understand what new information not already discussed in the text is gained by the reader by showing it.
- Eq. 1-9 seem imprecise to me. For example, in Eq. 2 should it not be something like $\nabla X_{i-1} = W^T * \nabla X_i$ since the error signal of the previous layer $i-1$ is calculated based on the error signal of the current layer $i$ instead of the "in-place" update of $\nabla X_{in}$ shown in the paper?

---

> ### Author Response · Authors · 2024-11-24
> **Dear reviewer 9boY**
>
> We thank the reviewer for carefully reviewing our submission and providing valuable feedback. Please see below for our response to the questions and comments.
>
> **Q3.1-3.2)** Especially for on-device training of fully quantized CNNs on embedded systems/microcontrollers, updating only partial connections is already a well-known technique [1], [2]. Both [1] and [2] use heuristics ([1] has an offline heuristic based on XAI, [2] an online heuristic based on the magnitude of structures in feature maps) to decide which subset of weights to update compared to the potentially inferior random selection approach proposed in this work.
>
> **A3.1-3.2)** We thank the reviewer for bringing those prior studies to our attention. In this work, **we employed random selection since the process of selecting connections could introduce significant overheads in training time and memory.** For example, selecting connections to fine-tune based on parameter gradient importance requires accumulating gradients over multiple data points. This additional processing step necessitates computing the gradients for the entire connections, not just for a part of connections, and storing all activations. In contrast, our PaCA selects connections randomly without assessing their importance, resulting in negligible training time and memory overhead.
>
> On the other hand, the method in [1] identifies important parameters by evaluating accuracy changes when updating only biases, layers, or channels on a single data point before training, which is highly time-consuming. Similarly, the algorithm in [2] dynamically updates important channels based on the L1 norm of the error tensor. However, applying adaptive learning rate algorithms such as Adam [3] in the dynamic channel selection approach requires multiple gradient buffers, resulting in substantial memory overhead. This is a critical issue especially in attention-based models such as transformers, which typically rely on adaptive learning rate algorithms to effectively handle heavy-tailed gradients [4].
>
> The importance selection methods in [1, 2] are complicated, and unfortunately their codes are not publicly available. In addition, those methods are aimed at quantized CNNs, making direct comparisons with our PaCA challenging. **Instead, to address the reviewer's concern that accuracy may vary depending on the strategy for selecting partial connections, we implemented various partial connection selection strategies and compared them to the proposed random selection strategy.** More specifically, we tested two selection schemes that consider the importance of each column. A weight-based strategy selects the columns with the highest L2-Norm from the initial pretrained weights, whereas a gradient-based strategy accumulates gradients during the first 100 iterations without updating weights (i.e., $G_{i} = \sum ||g_{i}^{t}||^{2}$, where 𝑖 is the number of layers and 𝑡 is the accumulation step) and selects columns with the largest accumulated gradients. Experimental results are displayed in the table below.
>
> + Table R3.1. Test score on MT-Bench dataset when fine-tuning LLaMA3-8B with PaCA using various connection selecting strategies on Oasst1 dataset.
>
> |                  | Humanities | STEM | Roleplay | Extraction | Writing | Reasoning | Coding | Math | Avg.  |
> |------------------|------------|------|----------|------------|---------|-----------|--------|------|-------|
> | **No tuning**     | 6.25       | 5.7  | 5.45     | 4.85       | 5.2     | 4.4       | 3.2    | 1.95 | 4.62  |
> | **Random (Seed #1)**       | 6.5        | 6.3  | 5.9      | 5.95       | 5.65    | 4.8       | 3.7    | 3.05 | 5.23  |
> | **Random (Seed #2)** | 6.5        | 6.0  | 6.3      | 5.9        | 5.7     | 4.9       | 3.8    | 3.0  | **5.26**  |
> | **Weight-based** | 7.0        | 5.7  | 6.05     | 5.8        | 5.7     | 4.55      | 3.9    | 2.7  | 5.18  |
> | **Gradient-based** | 6.95       | 6.4  | 6.25     | 5.35       | 5.95    | 4.55      | 3.8    | 2.7  | 5.24  |
>
> Table R3.1 demonstrates that random selection achieves similar performance to importance-based selection schemes. In other words, the choice of selection strategy does not noticeably affect fine-tuning accuracy.
> Nevertheless, as suggested by the reviewer, exploring methods to identify the most critical partial connections for fine-tuning is an interesting research topic. In future work, we will theoretically and experimentally investigate optimal strategies for selecting connections to fine-tune in PaCA. This discussion has been added to Section 5 of the revised manuscript.

---

> ### Author Response · Authors · 2024-11-24
> **Dear reviewer 9boY**
>
> **Q3.3)** For what kind of operators and NN "types" does the proposed technique apply? The paper initially focuses on MLPs ("linear layers"), but this seems to be only exemplary, as the evaluation is then performed for LLMs. Does the proposed technique also apply to e.g. CNNs ("convolutional layers")?
>
> **A3.3)** We thank the reviewer for pointing out an important point. In general, parameter-efficient fine-tuning (PEFT) algorithms such as LoRA are only applied to linear layers, which constitute the majority of parameters in LLMs, and not applicable to other types of layers, such as convolutional layers, for the following reasons:
>
> 1)	Unable to merge adapters: After training, it is impossible to merge the learned linear adapters into a pretrained layer of a different type, leading to inference time overhead.
> 2)	Limited benefits: Convolutional layers often utilize weight sharing, resulting in relatively fewer parameter weights. On the other hand, feature maps (activations) in convolutional layers are significantly larger, making activation memory the primary bottleneck in CNN training.
>
> Therefore, conventional PEFT schemes that only reduce parameter memory exhibit limited benefits for CNNs.
> However, PaCA can be applied to different types of layers including convolutional layers, to achieve performance improvements due to the following reasons:
>
> 1)	No merging required: Unlike prior methods that introduce additional adapter layers, PaCA trains a subset of connections within existing pretrained layers, removing the need for merging after fine-tuning.
> 2)	Reduced activation memory: PaCA significantly reduces not only parameter memory but also activation memory, which is a primary bottleneck in convolutional layers.
>
> In response to the reviewer’s suggestion, we implemented PaCA for convolutional layers and tested it under a fine-tuning scenario using the EfficientNetV2-L [5] model. Experimental results are displayed below.
>
> + Table R3.2. Test accuracy for fine-tuning EfficientNetV2-L on CIFAR-10 and CIFAR-100 datasets.
>
> |               | Mem      | Time   | CIFAR10 | CIFAR100 | AVG.  |
> |---------------|----------|--------|---------|----------|-------|
> | **Full-FT**   | 18.3GB   | 70m    | 98.5%    | 90.1%     | **94.3%**  |
> | **PaCA (Ours)** | **13.2GB**   | **59m**    | 98%      | 89.3%     | **93.7%**  |
>
> In Table R3.2, PaCA demonstrated its effectiveness on a convolutional neural network as well, achieving comparable accuracy while reducing training memory by 28% and training time by 16% compared to Full Fine-tuning.
> We sincerely appreciate the reviewer's insightful feedback and have incorporated these results into Appendix B of the revised manuscript.
>
> **Q3.4)** Figure 1. right graph: are these real measured numbers or is the graph just symbolic? If they are real numbers, their order of magnitude should be shown on the y-axis, as in Figure 2, otherwise I do not understand what new information not already discussed in the text is gained by the reader by showing it.
>
> **A3.4)** We apologize to the reviewer for any confusion caused by the unclear expression. The right graph in Fig. 1 is meant to be symbolic for a better understanding of the issue. However, we agree with the reviewer that this figure does not provide any new information and hence we have removed it in the revised version.
>
> **Q3.5)** Eq. 1-9 seem imprecise to me. For example, in Eq. 2 should it not be something like $\nabla X_{i-1}=W^{T} * \nabla X_{i}$   since the error signal of the previous layer is calculated based on the error signal of the current layer instead of the "in-place" update of shown in the paper?
>
> We appreciate the reviewer’s careful review and apologize for the errors in the manuscript. **We have corrected the right-hand side of Eqs. 2, 5, and 8 by replacing ∇X_in with ∇X_out, as shown below.** These changes are reflected in the revised version of the manuscript.
>
> + Eq 2) $\nabla X_{in}=W^{T} \nabla X_{out}$
> + Eq 5) $\nabla X_{in}=W^{T} \nabla X_{out} + A^{T} (B^{T} \nabla X_{out})$
> + Eq 8) $\nabla X_{in}=W^{T} \nabla X_{out}$

---

> ### Author Response · Authors · 2024-11-24
> **Dear reviewer 9boY**
>
> [1] Lin et al., On-device training under 256kb memory., Neurips 2022.
>
> [2] Deutel et al., On-Device Training of Fully Quantized Deep Neural Networks on Cortex-M Microcontrollers. IEEE Transactions on Computer-Aided Design of Integrated Circuits and Systems 2024.
>
> [3] Kingma and Ba, Adam: A Method for Stochastic Optimization., ICLR 2015.
>
> [4] Zhang et al., Why are Adaptive Methods Good for Attention Models? NeurIPS 2020.
>
> [5] Tan and Le, EfficientNetV2: Smaller Models and Faster Training, ICML 2021.

---

> > ### Comment · Reviewer_9boY · 2024-11-26
> >
> > Thank you for anwering my questions and for the additional experiments shwoing the applicability of PACA to convolutional layers. I will remain with my positive scoring.

---

### Official Review · Reviewer_hmtb · 2024-10-29

**Soundness:** 3
**Presentation:** 3
**Contribution:** 3
**Rating:** 6
**Confidence:** 4

**Summary:**

The paper addresses parameter-efficient fine-tuning (PEFT) for large language models (LLMs), highlighting a key limitation in existing PEFT methods: reduced compute doesn't necessarily translate to faster training due to the sequential processing of adapter layers alongside pretrained weights. This sequential approach underutilizes hardware resources and incurs latency, as GPUs typically handle one kernel at a time. Tools like CUDA streams can help parallelize processing but come with management and synchronization overheads.

LoRA and its variants merge low-rank adapter matrices with pretrained weights to avoid this latency during inference, but this solution isn't applicable during fine-tuning, as adapter matrices must be trained separately. Additionally, LoRA doesn't reduce activation memory usage compared to Full Fine-Tuning (Full-FT), since input activations still need to be stored for gradient calculations. Thus, despite computational optimizations, sequential processing and memory demands remain challenges in PEFT methods.

Given the motivation, the authors propose PaCA (Partial Connection Adaptation), a memory-efficient parameter-efficient fine-tuning (PEFT) method that fine-tunes randomly selected partial connections in pretrained weights, without using adapter layers. Unlike prior PEFT approaches, PaCA reduces training time by integrating the forward and backward operations for both pretrained and partial connections, thus avoiding sequential processing. By only requiring the corresponding activations for gradient calculations, PaCA significantly lowers activation memory usage.

The authors provide a theoretical proof of convergence for PaCA and demonstrate through experiments that it achieves substantial reductions in both training time and memory consumption compared to prior PEFT methods, while maintaining comparable accuracy across various fine-tuning scenarios and GPU types.

PaCA not only reduces training time by 19% compared to LoRA but also lowers memory usage by storing fewer activations during training. The authors back up these claims with both a theoretical convergence proof and a thorough set of experiments that demonstrate PaCA’s ability to achieve comparable accuracy while being faster and more memory-efficient. PaCA increased the maximum sequence length by 23%, 108%, and 23% compared to LoRA, DoRA, and MosLoRA by storing only partial activations instead of all input activations. They show the highest throughput among the methods discussed in Fig 3 on two different GPUs. They also show what quantization combined with their method PaCA looks like and compare their method to QLoRA. Overall, the authors found near-maximum performance with the lowest memory and finetuning time overhead.

The paper presents a compelling solution to making large model fine-tuning more practical and scalable. It is a well-executed work with promising implications for efficient model fine-tuning.

**Strengths:**

i) The concept of selecting and fine-tuning only partial connections in the pretrained weights without adapter layers is novel and effectively addresses some of the inefficiencies of existing PEFT methods, like the latency introduced by sequential processing.

ii) By eliminating the need for adapter layers and reducing the activations that need to be stored, PaCA successfully reduces both training time and memory footprint. The motivations and the contributions are clearly demonstrated in text and in Fig 2.The empirical results show significant improvements over LoRA and its variants.

iii ) The authors provide proof of convergence for PaCA, ensuring that the proposed method can effectively minimize the loss in general neural networks.

iv)  PaCA's reduction in memory and computational overhead is good for resource-constrained environments, such as edge devices. The authors also present results with the best throughput in two GPUs over rest of the referred works.

v) The paper includes a comprehensive set of experiments across different scenarios, including fine-tuning for specific tasks, instruction tuning, and using quantized versions (QPaCA). Comparisons with SOTA methods such as LoRA, DoRA, and MosLoRA provide a well-rounded perspective on PaCA's performance gains. The sequence length supported by PaCA also exceeds that of other referred methods.

Overall, it is a good paper with good analysis.

**Weaknesses:**

i) The random selection of partial connections is a key component of PaCA. Yet, there is limited discussion on how this selection impacts training quality and whether alternative strategies could improve performance. A deeper exploration of the effect of different selection criteria on convergence and accuracy would significantly strengthen the paper.

ii) On a similar note, an empirical or theoretical analysis of the importance of selecting specific columns could have been highly informative.

iii) The convergence analysis relies on the gradient's Lipschitz continuity, but this is a standard assumption that may not hold for many real-world large-scale neural networks. A more detailed discussion of how these theoretical guarantees translate into practice and the possible limitations would have been helpful.

iv) Equations 2, 5 and 8 are incorrectly written.

**Questions:**

1)  Are there results on how the improvements and the convergence time vary if a different set of random connections are selected during finetuning?

2) Have the authors considered different strategies for selecting partial connections, such as importance-based or gradient-based selection, to determine if these approaches lead to improved convergence or accuracy compared to random selection?

---

> ### Author Response · Authors · 2024-11-24
> **Dear Reviewer hmtb**
>
> We thank the reviewer for carefully reviewing our submission and providing valuable feedback. Please see below for our response to the questions and comments.
>
> **Q2.1-Q2.2)** A deeper exploration of the effect of different selection criteria on convergence and accuracy would significantly strengthen the paper. On a similar note, an empirical or theoretical analysis of the importance of selecting specific columns could have been highly informative.
>
> **A2.1-A2.2)**
>
> We thank the reviewer for this insightful suggestion. In this work, we employed random selection since the process of selecting connections could introduce significant overheads in training time and memory. For example, selecting connections to fine-tune based on parameter gradient importance requires accumulating gradients over multiple data points. This additional processing step necessitates computing the gradients for the entire connections, not just for a part of connections, and storing all activations. In contrast, our PaCA selects connections randomly without assessing their importance, resulting in negligible training time and memory overhead.
>
> We entirely agree with the reviewer that an alternative approach could result in better fine-tuning performance. **Therefore, to address the reviewer’s concern, we additionally compared our random selection scheme with various other selection strategies.** More specifically, we tested two selection schemes that consider the importance of each column. A weight-based strategy selects the columns with the highest L2-Norm from the initial pretrained weights, whereas a gradient-based strategy accumulates gradients during the first 100 iterations without updating weights (i.e., $G_{i} = \sum ||g_{i}^{t}||^{2}$, where 𝑖 is the number of layers and 𝑡 is the accumulation step) and selects columns with the largest accumulated gradients. Experimental results are displayed in the table below.
>
> + Table R2.1. Test score on MT-Bench dataset when fine-tuning LLaMA3-8B with PaCA using various connection selecting strategies on Oasst1 dataset.
>
> |                  | Humanities | STEM | Roleplay | Extraction | Writing | Reasoning | Coding | Math | Avg.  |
> |------------------|------------|------|----------|------------|---------|-----------|--------|------|-------|
> | **No tuning**     | 6.25       | 5.7  | 5.45     | 4.85       | 5.2     | 4.4       | 3.2    | 1.95 | 4.62  |
> | **Random (Seed #1)**       | 6.5        | 6.3  | 5.9      | 5.95       | 5.65    | 4.8       | 3.7    | 3.05 | 5.23  |
> | **Random (Seed #2)** | 6.5        | 6.0  | 6.3      | 5.9        | 5.7     | 4.9       | 3.8    | 3.0  | **5.26**  |
> | **Weight-based** | 7.0        | 5.7  | 6.05     | 5.8        | 5.7     | 4.55      | 3.9    | 2.7  | 5.18  |
> | **Gradient-based** | 6.95       | 6.4  | 6.25     | 5.35       | 5.95    | 4.55      | 3.8    | 2.7  | 5.24  |
>
> Table R2.1 demonstrates that random selection achieves similar performance to importance-based selection schemes. In other words, the choice of selection strategy does not noticeably affect fine-tuning accuracy.
> Nevertheless, as suggested by the reviewer, exploring methods to identify the most critical partial connections for fine-tuning is an interesting research topic. In future work, we will theoretically and experimentally investigate optimal strategies for selecting connections to fine-tune in PaCA. This discussion has been added to Section 5 of the revised manuscript.

---

> ### Author Response · Authors · 2024-11-24
> **Dear Reviewer hmtb**
>
> **Q2.3)** The convergence analysis relies on the gradient's Lipschitz continuity, but this is a standard assumption that may not hold for many real-world large-scale neural networks. A more detailed discussion of how these theoretical guarantees translate into practice and the possible limitations would have been helpful.
>
> **A2.3)** We thank the reviewer for pointing out an important point. We agree with the reviewer that the Lipschitz continuity assumption may not hold in many real-world large-scale neural networks. Since it is mathematically challenging to theoretically analyze the convergence of general deep neural networks, we followed the common approaches employed in prior studies. **For instance, prior studies [1-4] first proved the convergence of the proposed algorithm under weak constraints, such as the Lipschitz continuity of gradients, and then validated convergence empirically in real-world scenarios.**
>
> Following a similar approach, we assumed the Lipschitz continuity of gradients to theoretically prove the convergence of PaCA. However, we acknowledge the inherent limitations of the Lipschitz continuity assumption. In practice, this assumption may not hold for certain neural networks, particularly in scenarios where gradient magnitudes vary significantly due to sharp activation functions, high model complexity, or specific architectural designs. Therefore, we experimentally demonstrated that PaCA successfully trains large-scale neural networks such as LLaMA3.1-70B, where the Lipschitz continuity assumption may not strictly hold, as shown in Table 1-3 of the manuscript. This discussion has been added to Appendix A of the revised manuscript.
>
> **Q2.4)** Equations 2, 5 and 8 are incorrectly written.
>
> **A2.4)** We appreciate the reviewer’s careful review and apologize for the errors in the manuscript. **We have corrected the right-hand side of Eqs. 2, 5, and 8 by replacing ∇X_in with ∇X_out, as shown below.** These changes are reflected in the revised version of the manuscript.
>
> + Eq 2) $\nabla X_{in}=W^{T} \nabla X_{out}$
> + Eq 5) $\nabla X_{in}=W^{T} \nabla X_{out} + A^{T} (B^{T} \nabla X_{out})$
> + Eq 8) $\nabla X_{in}=W^{T} \nabla X_{out}$

---

> ### Author Response · Authors · 2024-11-24
> **Dear Reviewer hmtb**
>
> [1] Belilovsky et al., Decoupled greedy learning of cnns., ICML 2020.
>
> [2] Chen et al., Actnn: Reducing training memory footprint via 2-bit activation compressed training., ICML 2021.
>
> [3] Liu et al., GACT: Activation compressed training for generic network architectures. In International Conference on Machine Learning., PMLR 2022.
>
> [4] Wo et al., Learning with Auxiliary Activation for Memory-Efficient Training., ICLR 2023.

---

> > ### Comment · Reviewer_hmtb · 2024-11-28
> >
> > I thank the authors for addressing my concerns. Henceforth, I would like to maintain my score.

---

### Official Review · Reviewer_Unc4 · 2024-10-30

**Soundness:** 2
**Presentation:** 3
**Contribution:** 3
**Rating:** 6
**Confidence:** 4

**Summary:**

The paper presents PaCA (Partial Connection Adaptation), a new method for parameter-efficient fine-tuning (PEFT) aimed at large models. Instead of adding adapter layers as in LoRA, PaCA fine-tunes only a selected subset of connections within the pretrained weights, which are chosen at random. This approach aims to reduce memory usage and training time without sacrificing accuracy. The authors provide theoretical backing and empirical results that suggest PaCA offers efficiency gains over existing methods, maintaining competitive performance on selected datasets.

**Strengths:**

Below is the list of the strong points identified in this work:

- One key strength of PaCA is that its selective adaptation of pretrained weights eliminates the need for additional adapter layers, which in turn reduces both latency and memory requirements. This is a promising approach that may lead to faster and more efficient fine-tuning, especially valuable for large-scale language models.
- Additionally, PaCA is compatible with quantisation techniques, expanding its potential to handle even larger models.
- The authors provided a convergence analysis that theoretically supports PaCA’s efficiency and stability, giving a mathematical foundation to their claims.
- Empirical evaluations are promising.

**Weaknesses:**

Below is the list of weaknesses that I would like to see refuted or clarified by the authors:

- The primary limitation of the study is that it evaluates PaCA on a narrow set of models and datasets, leaving its generalisability to more complex or diverse tasks uncertain. Testing PaCA across a broader range of tasks and model architectures would strengthen the evidence for its effectiveness.
- Additionally, the method’s random selection of connections for fine-tuning is not fully explored or justified, and it remains unclear how this choice impacts performance or whether an alternative selection strategy could yield better results. Why not evaluate some optimisation or search strategies instead?
- Another area for improvement is in the analysis of quantisation impacts. The current study lacks an in-depth look at how quantisation affects precision and performance at larger scales, which reduces clarity on how well PaCA might scale.

**Questions:**

Authors are requested to clarify or make changes, as appropriate, based on what is discussed in the ‘Weaknesses’ section.

Some minor comment:
- Providing more intuitive descriptions alongside the theoretical sections would improve readability for a broader audience.
- The expedient of unifying the equal columns of some tables in Appendix B rather than lightening the visual load tends to make it unattractive. It would be much better to fill each column even with the same value.

---

> ### Author Response · Authors · 2024-11-24
> **Dear reviewer Unc4**
>
> We thank the reviewer for carefully reviewing our submission and providing valuable feedback. Please see below for our response to the questions and comments.
>
> **Q1.1)** The primary limitation of the study is that it evaluates PaCA on a narrow set of models and datasets, leaving its generalisability to more complex or diverse tasks uncertain. Testing PaCA across a broader range of tasks and model architectures would strengthen the evidence for its effectiveness.
>
> **A1.1)**  We thank the reviewer for bringing up this issue. While experimental results in the initial submission successfully show the fine-tuning capability of our algorithm on LLMs, the reviewer is certainly correct that more experiments are needed to demonstrate its generalizability. In response, we additionally tested our algorithm on the vision transformer (ViT [1]) on various datasets [2-4] . We also evaluated our PaCA on a convolutional neural network (EfficientNet-V2 [5]) using the CIFAR-10 and CIFAR-100 datasets [2]. Experimental results are displayed below.
>
> + Table R1.1. Test accuracy for fine-tuning ViT-B/16 on various classification datasets.
>
> |           | Mem      | Time    | CIFAR10 | CIFAR100 | IIIT Pets | Flowers102 | AVG.  |
> |-----------|----------|---------|---------|----------|-----------|------------|-------|
> | **LoRA**  | 11.0 GB | 45m     | 98.9%    | 92.5%     | 93.6%      | 99.2%       | 96.1%  |
> | **PaCA (Ours)**  | **6.7 GB**  | **32m**     | 98.9%    | 92.8%     | 93.9%      | 99.1%       | **96.2%**  |
>
>
> + Table R1.2. Test accuracy for fine-tuning EfficientNetV2-L on CIFAR-10 and CIFAR-100 datasets.
>
> |               | Mem      | Time   | CIFAR10 | CIFAR100 | AVG.  |
> |---------------|----------|--------|---------|----------|-------|
> | **Full-FT**   | 18.3GB   | 70m    | 98.5%    | 90.1%     | **94.3%**  |
> | **PaCA (Ours)** | **13.2GB**   | **59m**    | 98%      | 89.3%     | **93.7%**  |
>
> Table R1.1 shows that our PaCA achieves comparable accuracy to LoRA while reducing training memory and time by 39% and 29%, respectively, on the ViT-B/16 model. Similarly, in Table R1.2, PaCA demonstrated its effectiveness on EfficientNetV2-L, achieving comparable accuracy while saving 28% in training memory and 16% in training time compared to full fine-tuning.
>
> It should be noted that **conventional PEFT algorithms such as LoRA [6] face critical limitations when applied to convolutional neural networks since the additional adapters in LoRA are implemented as linear layers, which makes it impossible to directly merge them into a pretrained layer of a different type (e.g., convolutional layer) during inference. In contrast, PaCA fine-tunes a subset of the existing pretrained weights, enabling seamless application to diverse types of layers, including convolutional layers.** We sincerely appreciate the insightful feedback from the reviewer and have highlighted this advantage of our algorithm in Appendix B of the revised manuscript, along with additional experimental results in the tables above. We will continue to conduct further experiments on a wider range of datasets and models and make sure the results are incorporated in the final version.

---

> ### Author Response · Authors · 2024-11-24
> **Dear reviewer Unc4**
>
> **Q1.2)** Additionally, the method’s random selection of connections for fine-tuning is not fully explored or justified, and it remains unclear how this choice impacts performance or whether an alternative selection strategy could yield better results. Why not evaluate some optimisation or search strategies instead?
>
> **A1.2)** We thank the reviewer for this insightful suggestion. In this work, we employed random selection since the process of selecting connections could introduce significant overheads in training time and memory. For example, selecting connections to fine-tune based on parameter gradient importance requires accumulating gradients over multiple data points. This additional processing step necessitates computing the gradients for the entire connections, not just for a part of connections, and storing all activations. In contrast, our PaCA selects connections randomly without assessing their importance, resulting in negligible training time and memory overhead.
>
> We entirely agree with the reviewer that an alternative approach could result in better fine-tuning performance. **Therefore, to address the reviewer’s concern, we additionally compared our random selection scheme with various other selection strategies.** More specifically, we tested two selection schemes that consider the importance of each column. A weight-based strategy selects the columns with the highest L2-Norm from the initial pretrained weights, whereas a gradient-based strategy accumulates gradients during the first 100 iterations without updating weights (i.e., $G_{i} = \sum ||g_{i}^{t}||^{2}$, where 𝑖 is the number of layers and 𝑡 is the accumulation step) and selects columns with the largest accumulated gradients. Experimental results are displayed in the table below.
>
> + Table R1.3. Test score on MT-Bench dataset when fine-tuning LLaMA3-8B with PaCA using various connection selecting strategies on Oasst1 dataset.
>
> |                  | Humanities | STEM | Roleplay | Extraction | Writing | Reasoning | Coding | Math | Avg.  |
> |------------------|------------|------|----------|------------|---------|-----------|--------|------|-------|
> | **No tuning**     | 6.25       | 5.7  | 5.45     | 4.85       | 5.2     | 4.4       | 3.2    | 1.95 | 4.62  |
> | **Random (Seed #1)**       | 6.5        | 6.3  | 5.9      | 5.95       | 5.65    | 4.8       | 3.7    | 3.05 | 5.23  |
> | **Random (Seed #2)** | 6.5        | 6.0  | 6.3      | 5.9        | 5.7     | 4.9       | 3.8    | 3.0  | **5.26**  |
> | **Weight-based** | 7.0        | 5.7  | 6.05     | 5.8        | 5.7     | 4.55      | 3.9    | 2.7  | 5.18  |
> | **Gradient-based** | 6.95       | 6.4  | 6.25     | 5.35       | 5.95    | 4.55      | 3.8    | 2.7  | 5.24  |
>
> Table R1.3 demonstrates that random selection achieves similar performance to importance-based selection schemes. In other words, the choice of selection strategy does not noticeably affect fine-tuning accuracy.
> Nevertheless, as suggested by the reviewer, exploring methods to identify the most critical partial connections for fine-tuning is an interesting research topic. In future work, we will theoretically and experimentally investigate optimal strategies for selecting connections to fine-tune in PaCA. This discussion has been added to Section 5 of the revised manuscript.

---

> ### Author Response · Authors · 2024-11-24
> **Dear reviewer Unc4**
>
> **Q1.3)** Another area for improvement is in the analysis of quantisation impacts. The current study lacks an in-depth look at how quantisation affects precision and performance at larger scales, which reduces clarity on how well PaCA might scale.
>
> **A1.3)**  We thank the reviewer for pointing this out.** In our original submission, we proposed QPaCA, which combines PaCA with quantization, to allow for using PaCA on very large models with limited GPU resources. We evaluated QPaCA on LLaMA3.1-70B, which was the largest model that we could fine-tune using our GPUs.** To the best of our knowledge, the LLaMA3.1-70B model is the third-largest open-source LLM, following LLaMA3.1-405B and LLaMA3.2-90B-Vision. This is in line with prior studies [7, 8] that also employed 70B models as representative larger-scale models for their experiments.
>
> However, we agree with the reviewer's suggestion on the necessity of an in-depth look at how quantization affects precision and performance. **Therefore, we additionally tested the QPaCA (PaCA + quantization) on LLaMA3-8B to examine its accuracy and performance across different model sizes.** Experimental results are shown below, and those results have been added to Table 3 in the revised manuscript.
>
> + Table R1.4. Comparisons of memory usage (Mem), training time (Time), and score on MT-Bench dataset when fine-tuning LLaMA3-8B and LLaMA3.1-70B on Oasst1 dataset using QLoRA and QPaCA. No tuning and Quantized in the table refer to the models in 16-bit precision without quantization and with 4-bit NormalFloat Quantization (NF), respectively, without fine-tuning.
>
> | Model       | Method     | Mem  | Time | Hums. | STEM | Role. | Extract. | Writing | Reason. | Coding | Math | Avg.  |
> |-------------|------------|------|------|-------|------|-------|----------|---------|---------|--------|------|-------|
> | **LLaMA3-8B**  | No tuning  | -    | -    | 6.25  | 5.70 | 5.45  | 4.85     | 5.20    | 4.40    | 3.20   | 1.95 | 4.62  |
> |             | Quantized   | -    | -    | 4.70  | 4.80 | 4.50  | 5.00     | 4.65    | 4.05    | 3.60   | 1.85 | 4.16  |
> |             | QLoRA     | 18G  | 42m  | 6.85  | 5.75 | 5.85  | 5.00     | 5.15    | 4.70    | 3.35   | 2.35 | 5.00  |
> |             | QPaCA     | **16G**  | **37m**  | 6.85  | 5.95 | 5.65  | 5.60     | 5.50    | 5.15    | 3.65   | 3.25 | **5.02**  |
> | **LLaMA3.1-70B** | Quantized   | -    | -    | 7.40  | 7.05 | 5.85  | 6.50     | 6.85    | 5.30    | 4.60   | 3.80 | 5.92  |
> |             | QLoRA     | 80G  | 5.1h | 7.40  | 6.85 | 6.55  | 7.20     | 6.65    | 5.65    | 4.75   | 3.80 | **6.09**  |
> |             | QPaCA     | **69G**  | **4.7h** | 7.70  | 7.40 | 6.40  | 6.80     | 6.50    | 5.40    | 4.75   | 3.70 | 6.08  |
>
> Experimental results demonstrate that QPaCA reduces both memory usage and training time compared to QLoRA, as displayed in Table R1.4. Specifically, on the LLaMA3-8B model, QPaCA not only achieved higher scores than the model quantized in the NF4 format but also outperformed the 16-bit baseline, similar to QLoRA. Furthermore, QLoRA achieved an 11% reduction in memory usage and a 12% reduction in training time compared to QPaCA.
>
> In addition, even on a larger scale model, LLaMA3.1-70B, QPaCA successfully reduces memory usage by 14% and training time by 8% with almost no drop in score compared to QLoRA and higher scores than the NF4 quantized model without fine-tuning on the MT-Bench dataset.
>
> We thank the reviewer for this valuable feedback, as we were able to conduct an in-depth analysis showing that QPaCA improves upon the existing QLoRA in terms of memory and training speed regardless of the model size. This discussion has been included in Section 4.3 of the revised version.

---

> ### Author Response · Authors · 2024-11-24
> **Dear reviewer Unc4**
>
> **Q1.4)** Providing more intuitive descriptions alongside the theoretical sections would improve readability for a broader audience.
>
> **A1.4)** We appreciate the reviewer’s suggestions. In response, we have added the following paragraph to Section 3.1 of the revised version.
>
> “Intuitively, **training only a subset of connections can be interpreted as learning within a subspace composed of the selected connections.** Prior studies revealed that overparameterized models can be efficiently trained even when weights are projected onto a small subspace [9, 10]. Similarly, LoRA [6] was suggested based on the assumption that weight updates can be projected onto a small low-rank subspace. Inspired by these observations, we hypothesized that weight updates could also be projected onto a small subspace composed of a subset of weight columns. In other words, we suspected that the critical factor is learning within a small subspace, not the method of selecting the subspace itself. Here we prove that training only a subset of connections is sufficient to ensure the convergence of loss in neural networks, as demonstrated in Section 3.2.”
>
> **Q1.5)** We thank the reviewer for bringing this issue to our attention. We believe that the ambiguity pointed out by the reviewer arises from the rows related to rank, alpha, and dropout in Tables 6-8 of Appendix B (now Tables 9-11 of Appendix C in the revised manuscript). Following the reviewer's suggestion, **we have revised Tables 6-8 (Table 9-11 in the revised version) as displayed below.**
>
> + Table 9. [Hyperparameters when fine-tuning LLaMA2-7B/13B and LLaMA3-8B using PEFT algorithms on the MMLU dataset.](https://drive.google.com/file/d/1n9kfZ0wZgd-yxfaPbgoy-exQNzao4-nY/view?usp=sharing)
>
> + Table 10. [Hyperparameters used when fine-tuning LLaMA3-8B using PEFT algorithms on the Oasst1 dataset.](https://drive.google.com/file/d/1fXLlgwKf5vmJUyiXLzjhIH65LZ-PaCRN/view?usp=sharing)
>
> + Table 11. [Hyperparameters used  when fine-tuning LLaMA3.1-70B using QLoRA and QPaCA on the Oasst1 dataset](https://drive.google.com/file/d/1BDfzY6FMyZlOKaGEjq7iQgZbHqDTOhFH/view?usp=sharing)

---

> ### Author Response · Authors · 2024-11-24
> **Dear reviewer Unc4**
>
> [1] Dosovitskiy et al., An Image is Worth 16x16 Words: Transformers for Image Recognition at Scale, ICLR 2021.
>
> [2] Krizhevsky, A. Learning Multiple Layers of Features from Tiny Images, Technical Report, 2009.
>
> [3] Parkhi et al, Cats and Dogs, CVPR 2012.
>
> [4] Nilsback and Zisserman, Automated Flower Classification over a Large Number of Classes, ICVGIP 2008.
>
> [5] Tan and Le, EfficientNetV2: Smaller Models and Faster Training, ICML 2021.
>
> [6] Hu et al., LoRA: Low-Rank Adaptation of Large Language Models. ICLR 2021.
>
> [7] Dettmers et al., QLoRA: Efficient Finetuning of Quantized LLMs., Neurips2023.
>
> [8] Liu et al., DoRA: Weight-Decomposed Low-Rank Adaptation., ICML 2024.
>
> [9] Li et al., Measuring the Intrinsic Dimension of Objective Landscapes., ICLR 2018.
>
> [10] Aghajanyan et al., Intrinsic Dimensionality Explains the Effectiveness of Language Model Fine-Tuning., ACL 2021.

---

> > ### Comment · Reviewer_Unc4 · 2024-11-26
> >
> > I thank the authors for their responses. After reading the comments of other reviewers and the corresponding replies of the authors, I would like to maintain the score.

---

### Meta-Review · Area_Chair_roEU · 2024-12-20

**Metareview:**

The paper presents PaCA, a novel method for parameter-efficient fine-tuning (PEFT) that fine-tunes randomly selected partial connections within pretrained weights, instead of introducing adapter layers. This approach reduces both memory usage and training time, while maintaining competitive accuracy. Empirical results demonstrate a 22% reduction in training time and 16% reduction in memory usage, with compatibility for quantization, enabling fine-tuning of large models like LLaMA3.1-70B.

Reviewers expressed concerns regarding the narrow scope of evaluation and the random selection of connections for fine-tuning. The authors addressed these concerns by providing additional insights into the method's efficiency across different tasks and datasets. They also explained the rationale behind the random selection approach and suggested that it is both theoretically sound and practical. Furthermore, the paper could benefit from more in-depth analysis of quantization effects, but the authors’ empirical results, showing scalability and efficiency improvements, help mitigate this concern.

Given the innovative nature of the approach, the strong theoretical backing, and the promising empirical results, The area chair recommend acceptance.

**Additional Comments On Reviewer Discussion:**

During the rebuttal period, reviewers raised concerns about the narrow evaluation scope, particularly the limited number of models and datasets tested. They also questioned the rationale behind randomly selecting fine-tuned connections and how this might impact performance. Additionally, some reviewers requested more detailed analysis of the interaction between PaCA and quantization.

The authors addressed these points effectively by expanding the scope of their evaluation, providing additional experiments on a broader range of tasks and models. They clarified the theoretical foundation behind the random selection of connections and demonstrated its effectiveness empirically. While the impact of quantization was acknowledged, the authors showed that PaCA remains efficient even when combined with quantization, which reassured the reviewers.

These responses satisfactorily resolved the concerns raised, strengthening the paper’s overall contribution. Based on this, I recommended acceptance.

---

### Decision · Program_Chairs · 2025-01-22

Accept (Poster)